# Proactive Agents for Multi-Turn Text-to-Image Generation Under Uncertainty

**Meera Hahn** [* 1]   **Wenjun Zeng** [1]   **Nithish Kannen** [1]   **Rich Galt** [1]   **Kartikeya Badola** [1]   **Been Kim** [1]   **Zi Wang** [* 1]

## Abstract

User prompts for generative AI models are often underspecified, leading to a misalignment between the user intent and models' understanding. As a result, users commonly have to painstakingly refine their prompts. We study this alignment problem in text-to-image (T2I) generation and propose a prototype for proactive T2I agents equipped with an interface to (1) actively ask clarification questions when uncertain, and (2) present their uncertainty about user intent as an understandable and editable *belief graph*. We build simple prototypes for such agents and propose a new scalable and automated evaluation approach using two agents, one with a ground truth intent (an image) while the other tries to ask as few questions as possible to align with the ground truth. We experiment over three image-text datasets: ImageInWords (Garg et al., 2024), COCO (Lin et al., 2014) and DesignBench, a benchmark we curated with strong artistic and design elements. Experiments over the three datasets demonstrate the proposed T2I agents' ability to ask informative questions and elicit crucial information to achieve successful alignment with at least 2 times higher VQAScore (Lin et al., 2024) than the standard T2I generation. Moreover, we conducted human studies and observed that at least 90% of human subjects found these agents and their belief graphs helpful for their T2I workflow, highlighting the effectiveness of our approach. Code and DesignBench can be found at https://github.com/google-deepmind/proactive_t2i_agents.

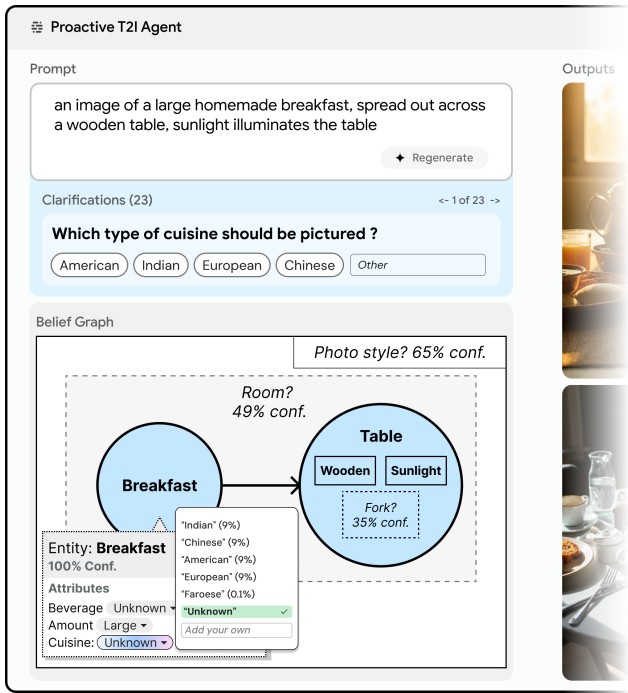

*Figure 1.* Our proactive T2I agent clarifies a user prompt with questions, incorporates user feedback, and expresses its uncertainty and understanding as an editable belief graph.

## 1. Introduction

A fundamental challenge in the development of AI agents is how to foster effective and efficient multi-turn communication and collaboration with human users to achieve user-defined goals, especially when faced with the common issue of vague or incomplete instructions from humans.

As a testbed for the communication problem, this work uses text-to-image (T2I) generation, where recent advancements (Baldridge et al., 2024; Betker et al., 2023; Podell et al., 2023; Yu et al., 2023) have enabled the creation of stunning images from complex text descriptions. T2I exemplifies the challenge of moving from low-information (prompts) to high-information (images), forcing models to make implicit assumptions to resolve prompt uncertainty. Moreover, T2I offers well-defined metrics and a clear problem scope, enabling comprehensive automatic evaluation.

Interpretations of prompts can vary significantly across di-

[*]Equal contribution  [1]Google DeepMind. Correspondence to: Zi Wang <wangzi@google.com>.

*Proceedings of the 42^{nd} International Conference on Machine Learning*, Vancouver, Canada. PMLR 267, 2025. Copyright 2025 by the author(s).

verse populations, groups, and cultures, resulting in misalignment between user intents and model outputs. For example, the prompt "A breakfast plate" can evoke a wide range of mental images depending on cultural background. While a typical T2I model might generate a generic breakfast plate, it is unlikely to produce a culturally appropriate cuisine without further information about the user's background and desired cuisine. These model assumptions can lead to user frustration and dissatisfaction, repeated prompt adjustments, while also potentially propagating bias (Kannen et al., 2024; Basu et al., 2023) and causing offense (Bianchi et al., 2023).

Beyond cultural contexts and differences, similar challenges arise with complex scenes. Consider the prompt "Image of a rabbit near a cat." Standard T2I models will likely generate an image containing both animals, however, it is unlikely the generated images will capture the specific details and spatial relationships the user envisions. These combined factors often result in a frustrating cycle of trial and error, as users repeatedly refine their prompts in an attempt to guide the model toward the desired output (Vodrahalli & Zou, 2024; Huang et al., 2024; Sun & Guo, 2023).

In this paper we pursue a quest for agency in T2I generation with the aim to mitigate the issues that arise from passive T2I models. In the proposed setup, T2I agents actively engage with human users to provide a collaborative and interactive experience for image creation. We envision that these T2I agents will be able to (1) express and visualize their beliefs and uncertainty about user intents, (2) allow human users to directly control agents' beliefs beyond just text descriptions, and (3) proactively seek clarification from the human user to iteratively align their understanding with what the human user intends to generate.

At the core of these agents, we build in a graph-based symbolic belief state, termed the **belief graph**, that allows agents to express uncertainty about entities (e.g., "breakfast", "table") that may appear in the image, attributes of entities (e.g., cuisine type) and relations between entities (e.g., "breakfast" being on "table"). Given a prompt, we use an LLM and constrain its generation to adhere to the belief graph data structure, which includes probability estimates for entity presence and potential values for attributes and relations.

For example, for a prompt related to "homemade breakfast on a wooden table" in Figure 1, the agent's belief graph might indicate a 9% probability that the cuisine is American, while also recognizing other possibilities like Indian or Faroese. The belief graph also anticipates entities not explicitly mentioned but the user might want. Even though the user only specifies "homemade breakfast", the agent might be 35% confident that a fork will be in the generated images. The expression of uncertainty enables the agent to inspire new ideas and to guide users in editing uncertain

items within the graph.

However, uncertainty alone is an insufficient trigger for clarification, as not all uncertain elements are equally consequential. For instance, the breakfast's cuisine type is likely more impactful on the generated images than the table's wood type. To address this, we embed importance scores alongside probabilities within the belief graph for each entity, attribute, and relation.

Based on both uncertainty and importance, the agent proactively asks clarification questions. It can, for instance, identify the most uncertain attribute of an important entity (e.g., cuisine type of "breakfast"). An LLM is then used to formulate a relevant question (e.g., "Which type of cuisine should be pictured?"). To make answering easier, the agent provides likely options for users to select from (e.g., American, Indian, European, Chinese), though users can also type their own answers directly.

Conditioned on user answers and edits in the belief graph, the agent uses an LLM to update the prompt and transitions to a new belief graph by modifying the uncertainty of items clarified by the user. With the updated prompt, the agent calls an off-the-shelf T2I model to generate images.

The structure of our agent prototypes is highly modular, making it easy to improve each component individually, e.g., changing the strategy for asking questions, updating the belief graph construction method, and switching to better LLMs or T2I models when they become available.

To address the lack of evaluation standards for proactive T2I agents, we develop a scalable and automatic **evaluation pipeline** with a collection of metrics. The system first constructs a simulated user whose ground truth intent is a prefixed image, and translates the image to a short, underspecified text description. We then let a proactive agent and the simulated user play a multi-turn question answering game. The proactive agent aims to collect as much information as possible to generate images close to the ground truth, while the simulated user provides concise answers to agent questions. We then use text and image similarity metrics to evaluate the generated image against the goal image.

To better represent intents of diverse artists and designers (a key group of T2I users), we have created **DesignBench**, a carefully hand-curated benchmark. This benchmark contains aesthetic scenes with multiple entities and interactions between them. It also provides both short and long captions that standardize simulated users in our evaluation pipeline. DesignBench features diversity between photo-realism, animation and multiple styles, allowing a robust testing with the use case of artists and designers in mind.

We run both automatic and human evaluations on DesignBench, the COCO dataset (Lin et al., 2014) and ImageIn-

Words (Garg et al., 2024). Our proposed agents achieve at least 2 times higher VQAScore (Lin et al., 2024) than traditional single-turn T2I generation within just 5 turns of interaction. Additionally we conduct a human user study in which over 90% human subjects expect proactive clarifications to be helpful, about 85% find belief graphs helpful, and 58% think the question asking feature of agents could deliver value to their work very soon, or immediately. Participants prefer images generated by our agents over those of single-turn T2I systems in more than 80% cases of the 550 prompt-image pairs used in the human study. This shows that our simple framework, which uses an off the shelf model without fine-tuning, can provide significant gains.

We believe our approach, where agents actively gather information from the user to reduce uncertainty, will empower a diverse range of users to leverage generative models for their needs and make these models safer and more responsible. By enabling direct control via interaction and understandable belief graphs, we move beyond prompt crafting and towards a future where tools like T2I are more accessible to people without special knowledge of writing prompts.

## 2. Background

The belief graph in our work is closely related to symbolic world representations. See §B for other related works on human-computer interaction, interpretability, etc.

**World states.** Classical AI represents the world with symbols (McCarthy & Hayes, 1969; Minsky, 1974; 1988; Pasula et al., 2007; Kaelbling & Lozano-Pérez, 2011). For example, in the blocks world (Ginsberg & Smith, 1988; Gupta & Nau, 1992; Alkhazraji et al., 2020), a state can be $is\_block(a) \land is\_red(a) \land on\_table(a) \land is\_block(b) \land is\_blue(b) \land on(b, a)$, describing that there are a red block and a blue block, referred to as $a$ and $b$, block $a$ is on a table, and block $b$ is on $a$. Such world states include **entities** (e.g., $a$ and $b$), their **attributes** (e.g., position $on\_table$, characteristics $is\_block$) and **relations** (e.g., $on(b, a)$) which are critical for enabling a robot to know and act in the world. The predicates like $on\_table$ are pre-defined and hardcoded into robot systems.

In linguistics, Davidson (1965; 1967b;a) introduce logic-based formalisms of meanings of sentences. The semantics of a sentence is decomposed to a set of atomic propositions, such that no propositions can be added or removed from the set to represent the meaning of the sentence. Cho et al. (2023) propose Davidsonian Scene Graph (DSG) which represents an image description as a set of atomic propositions (and corresponding questions about each proposition) to evaluate T2I alignment.

We borrow the same concept as symbolic world representations and scene graphs, except that the agent needs to

represent an imaginary world. The image generation problem can be viewed as taking a picture of the imaginary world. The world state should include all entities that are in the picture, together with their attributes and relations.

**Belief states.** Term "belief state" (Nilsson, 1986; Kaelbling et al., 1998) has been used to describe a distribution over states. E.g., for block $a$, we might have $p(on\_table(a)) = 0.5$ and $p(\neg on\_table(a)) = 0.5$, which means the agent is unsure whether the block is on a table. To represent the T2I agent's belief on which image to generate, we need to consider the distribution over all possible "worlds" in which the picture can be taken. This distribution can be described by the probabilities that an entity appears in the picture, an attribute gets assigned a certain value, etc.

In contrast to classical belief states, our belief graphs incorporate importance scores tailored to generative AI tasks and leverage LLMs to automatically derive predicates from user prompts. This approach makes the belief graphs highly adaptable to new tasks, eliminating the need for the rigid, hardcoded rules in traditional AI systems.

## 3. Belief Graphs

Instead of using hardcoded symbols in classic belief representations (Fikes & Nilsson, 1971) described in §2, We adopt a simple approach: leveraging LLMs as a tool to adaptively generate belief graphs from image descriptions of any length. While more sophisticated approaches, such as those using activations, can be explored as future work, this simple method allows for flexible belief graph generation across many types of images.

In a belief graph, nodes are entities and edges are relations between entities. Each entity is composed of its name, description, type, probability, importance score and attributes.

We have 3 **types of entities**: (a) *explicit*: entities mentioned in the prompt, (b) *implicit*: entities not mentioned in the prompt but likely to appear, e.g., "fork" in Figure 1, and (c) *background entities*, such as *image style, time of day, location*, which play important roles in constructing the image.

Each entity's **probability** reflects its likelihood of appearing in the generated image, given the user's prompt. This can be thought of as the output of a classifier. Explicit and background entities typically have probabilities close to 1 or 0, clearly indicating their presence or absence. Implicit entities, however, usually have probabilities between 0 and 1, reflecting their uncertain inclusion.

An entity's **importance score** (normalized between 0 and 1) reflects its relevance and impact on the image. For example, in the "breakfast spread out across a wooden table" prompt in Figure 1, we care more about the specific breakfast items

than the type of wood the table is made of. Therefore, "breakfast" should have a higher importance score than "table". It is worth noting that different people have different judgments on the relative importance of elements in an image. In this work, we make a simplifying assumption that the agent doesn't consider the distribution over people's perceptions of importance scores.

An entity has multiple **attributes**, each with a name, a list of value-probability pairs (e.g., see the *cuisine* attribute of "breakfast" in Figure 1) and an importance score. An attribute's importance score shows its relevance to the image. For example, the cuisine of the breakfast might be more important than its exact size.

**Relations** are a special case of attribute that connects two entities, along with other information that an attribute can have. For each relation, the values in the list of value-probability pairs can be "part of", "under", "overlap", etc., describing the spatial relations between the two entities. Besides explicit relations (e.g., breakfast *on* table), belief graph also includes potential implicit relations. For instance, the *spatial relation* between "fork" and "table" might have a name like "in the center of".

## 4. Proactive T2I Agent Design

We provide high-level principles and design that guide our agent how to behave and interact with users to generate desired images from text through multi-turn interactions. The goal of the agent is to generate images that match the user's intended image as closely as possible with minimal back-and-forth, particularly in cases with underspecified prompts and the agent needs to gather information proactively. This requires a decision strategy on information gathering to trade off between the cost of interactions and the quality of generated images. The formal problem definition can be found in §C. Algorithm 1 summarizes how the agent constructs an initial belief graph from a user prompt, takes actions (including asking questions and presenting belief graphs), incorporates user feedback during interaction, and updates the belief graph and prompt.

---

**Algorithm 1** Belief parsing and interaction

---

1: **Input:** Initial user prompt ($p$)
2: **Initialization:**
3: Parse entities from prompt $p$ (E.5)
4: Parse attributes and relations from entities and $p$ (E.6, E.7)
5: Construct initial belief graph $b$
6: **for** $turn \leftarrow 1$ **to** $max\_turn$ **do**
7:     Take action $a \leftarrow \pi(b, p)$
8:     Observe user feedback $o$
9:     Update $b, p \leftarrow \tau(b, p, o)$ (E.9)
10: **end for**

---

### 4.1. Belief Graph Transitions

The agent's belief graph gets updated when the agent receives new information through user feedback. This feedback can come from agent-user dialog or through a user's direct manipulation of the belief graph on the agent interface (Figure 1). The transition process integrates information from the initial user prompt, the conversation history, interaction and the previous belief. In an agent prototype, we use a simple approach to implement the transition: Generate a comprehensive prompt that summarizes all interactions and information gathered thus far. This merged prompt is then used to re-generate the belief, effectively incorporating the new information into a refreshed representation. The implementation details can be found in §E.4.

### 4.2. Asking Questions to Gather Information

We identify the following principles for an agent to ask the user questions about the underspecified prompt and their intended image: (i) **No Redundancy**: The question should not collect information present in the history of interactions with the user. (ii) **Uncertainty Reduction**: The question should aim to reduce the agent's overall uncertainty about what contents should be included in the generated image, e.g., entities, attributes, spatial layout, style. (iii) **Relevance**: The question should be based on the user prompt. (iv) **Easy-to-Answer**: The question should be as concise and direct as possible to ensure it is not too difficult for the user to answer. The No Redundancy and Relevance principles are self-explanatory, we detail the other principles below.

**The Uncertainty Reduction principle** aims to let the agent elicit information about various characteristics of the desired image, which the agent is unsure of.

First, the agent needs to know what characteristics of images are important. For the "Image of a rabbit near a cat" prompt, some examples include: (i) Attributes of the subjects, such as breed, size, or color, with questions like *What kind of rabbit? What color is the cat?*; (ii) Spatial relationships between the subjects, such as proximity and relative position (*Are the rabbit and cat close to each other? Are they facing each other?*); (iii) Background information, such as location, style and time of day (*Are they in a park or at home?*); and (iv) Implicit entities that might not be explicitly mentioned in the initial prompt but are relevant to the user's vision (*Are there any other animals or people present?*).

Second, the agent needs to know its own uncertainty about those characteristics. In the belief graph of the agent, the uncertainty is explicit. One strategy is to form questions about the image characteristics that the agent is most uncertain about. We can define an acquisition function that takes into account the entropy over entities, attributes, relations, as well as the importance scores. We discuss more in §E.2.

Third, the agent needs to update its own uncertainty once the user gives a response to its question (a.k.a. transition in §4.1). Then, it can construct questions again based on its updated uncertainty estimates. This iterative clarification process allows the agent to progressively refine its understanding of the user's intent and generate an image that more accurately reflects their desired output.

**The Easy-to-Answer principle** aims to reduce the mental effort required from users when responding to questions. One strategy is for the agent to offer pre-defined answer choices. These choices are typically the ones the agent deems most probable. For example, *What color is the cat? (a) Black (b) Brown (c) Orange (d) Other (please specify).*

## 5. Implemented Agent Prototypes

We propose and experiment with three agent prototypes (Ag1, Ag2, and Ag3). All agents use the same belief graph construction approach in Algorithm 1. They differ in how they address the four question-asking principles (detailed in §4.2), specifically through their decision strategy $\pi$.

### 5.1. Ag1: Heuristic Score Agent

Ag1 asks questions about an entity's existence, attribute values, or relation types when these elements are both highly important and uncertain. It calculates a heuristic score in Equation (1) by multiplying importance scores and entropy of every belief graph element, then selects the highest-scoring question. We then prompt an LLM (§E.12) to craft easy-to-answer, multiple-choice questions like, "What color of the rabbit do you have in mind? a. black, b. white, c. brown, d. unknown, or something else?"

For belief graph transition, Ag1 re-generates the belief graph based on the merged prompt ($p$ in Line 9 of Algorithm 1), and applies the post-processing logic outlined in §E.4 to ensure consistency and *prevent redundancy*.

Figure 4 shows Ag1 excels at constructing easy-to-answer questions. However, its hardcoded post-processing and heuristic function doesn't consistently prevent redundancy, largely due to potential LLM errors in parsing beliefs from prompts. For example in Figure 5, Ag1 asks whether "cake" exists, even when it was mentioned in the initial prompt. This can happen if the LLM assigns "cake" a less than 100% existence probability and a high importance score.

### 5.2. Ag2: Belief-prompted Agent

Ag2 employs a belief-based $AICQ_B$ prompt in §E.11: instead of using a heuristic score like Ag1, it directly prompts an LLM to produce a question. The LLM generates each question using the belief graph, merged prompt, and conversation history. This comprehensive input allows it to craft highly relevant, informative and non-redundant queries. The

LLM is also guided with specific instructions on how to construct ideal questions, such as "The question should be as concise and direct as possible," to make questions less difficult to answer, however they are not limited to multiple choice. Questions are often of the structure "What color is the rabbit in the image?"

Ag2 does not require the post-processing logic (§E.4) for the belief transition, since the LLM is expected to use the conversation history to judge redundancy. Our experiments in §6.2 support this hypothesis, showing that Ag2 tends to achieve better performance than Ag1 with fewer turns of questions. However, Ag2 generates fewer easy-to-answer questions than Ag1, as shown in Figure 4.

### 5.3. Ag3: Principle-prompted Agent

The primary distinction between Ag2 and Ag3 is the information they are prompted with. While Ag2 leverages a belief graph and merged prompt, Ag3 uses the $AICQ_{base}$ prompt (§E.10), which exclusively relies on the conversation history and the question-asking principles from §4.2.[1]

Ag3 tends to ask more open-ended questions such as "What else is present in the image?" and more complex questions like "Is the cake on a plate or a stand, and what is its color and shape?" Figure 4, Figure 3 and Table 1 show that these types of questions are more difficult for users to answer but gain more information about user intent.

## 6. Experiments

We study the effectiveness of the agent design via both **automatic evaluation**, which uses a simulated user to converse with a T2I agent, and **human study**, which studies the efficacy of our framework with human subjects.

### 6.1. Automatic Evaluation

We simulate user-agent conversations using LLM self-play in Algorithm 2, where the conversation begins with a goal image and a detailed ground truth prompt describing it. We use the algorithm similar to *Ag2* to simulate the user, answering questions based on the ground truth prompt and the belief graph generated from the ground truth prompt. More details of the simulated user can be found in §E.3. We run each agent-user conversation for 15 turns and compute various metrics at the end of each turn. Figure 2 part b shows the multi-turn set up that we use in our results.

6.1.1. SETUPS FOR AGENTS AND BASELINE

**Baselines.** We use a standard T2I model as a baseline, which directly generates an image based on a prompt without asking any questions. We refer to this baseline as 'T2I'.

---

[1]We tried prompting an LLM with only history or simple instructions like "ask a question", but the quality of the clarification questions was noticeably worse than prompting with the principles.

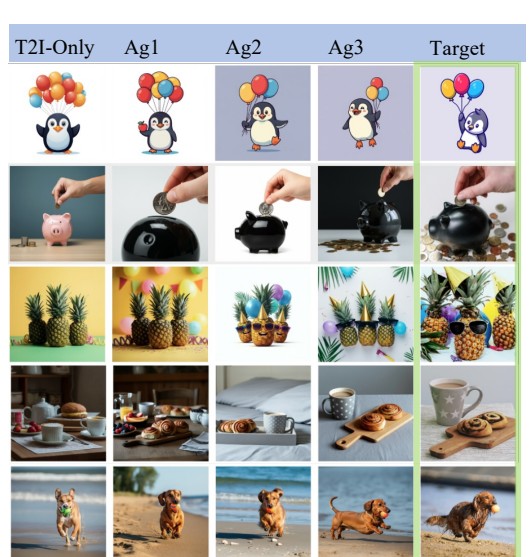

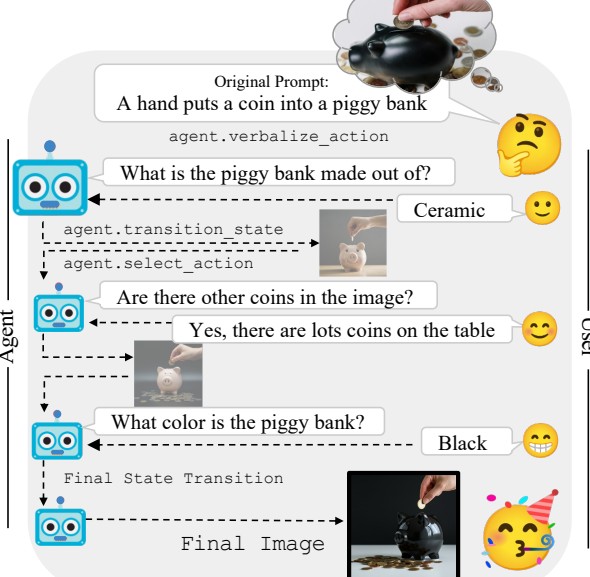

a) generated outputs and target image       b) multi-turn Ag3 example – real generated outputs

*Figure 2.* **a)** Each column displays the output of an agent after 15 turns - the right most column shows target image, which belongs to DesignBench. **b)** A visualization of the multi-turn evaluation set up in the experiments. These are real generated outputs and simulated user outputs at turns 3, 10 and 15.

**Agents.** We use Ag1, Ag2 and Ag3 with question-asking strategies introduced in §5. Further implementation details of each agent can be found in §E.

**Model Selection.** We implement the agent belief parsing and interaction in Algorithm 1 on top of the Gemini 1.5 (Gemini Team Google, 2024) using the default temperature and a 32K context length. For T2I generation, we use Imagen 3 (Baldridge et al., 2024) across all baselines given its recency and prompt-following capabilities. We used both the models served publicly using the Vertex API.[2]

### 6.1.2. DATASETS.

The evaluation data consists of tuples: $(\mathbf{I}, p_0, c, b_{gt})$, where $\mathbf{I}$ represents the target image, $p_0$ is an initial prompt describing only the primary elements of the scene, $c$ is a ground truth caption providing a detailed description of $\mathbf{I}$, including spatial layout, background elements, and style, and $b_{gt}$ is the ground truth belief graph constructed via parsing $c$. The initial prompt $p_0$ is intentionally less detailed than $c$ to necessitate multi-turn refinement. The evaluation framework we propose directly assesses the agent's ability to guide the user towards the target image $\mathbf{I}$ starting from a $p_0$.

Existing image-caption datasets primarily focus on simple scenes (Deng et al., 2009; Krizhevsky et al., 2009; Deng, 2012) or focus on very specific categories (Liu et al., 2016; Liao et al., 2022). With the aim for complex realistic images

for testing the robustness of the agents, we evaluate over the Coco-Captions dataset validation split (Chen et al., 2015) and ImageInWords (Garg et al., 2024) dataset. ImageIn-Words dataset contains a diverse set of realistic and cartoon images and has human annotators create dense detailed captions that describe attributes and relationships between objects in the image. In ImageInWords evaluations we use the long human annotation as the ground truth caption and auto-summarize the long caption with Gemini 1.5 Pro to create *starting prompt* $p_0$. For each image in Coco-Captions, the dataset provides multiple human generated captions which are short and describe the basic elements and the interactions in the image. We select the shortest of captions per image as the *starting prompt* $p_0$. We obtain the ground truth caption by expanding $p_0$ via prompting Gemini 1.5 Pro with ground truth image and asking it to add more details of the attributes of the entities in the image as well as the style and image composition.

COCO-Captions and ImageInWords datasets lack artistic and non-photorealistic imagery often desired by designers and artists. To better evaluate our target for flexible use cases such as by artists, we introduce and release **DesignBench**[3], a novel dataset comprising 30 scenes specifically designed for this purpose. Each scene follows the $(\mathbf{I}, p_0, c, b_{gt})$ format described earlier. DesignBench includes a mix of human generated cartoon graphics, photorealistic yet improbable

---

[2]https://cloud.google.com/vertex-ai

[3]https://huggingface.co/datasets/meerahahn/DesignBench

scenes, and artistic photographic images. Initial prompt $p_0$ and ground truth caption $c$ in DesignBench are generated via prompting Gemini 1.5 Pro with ground truth image.

We validate the quality of the Gemini 1.5 Pro generated captions, $c$, by performing Text to Image (VQA) Similarity between $c$ and the $\mathbf{I}$. For DesignBench the mean T2I VQA similarity between the ground truth caption and ground truth image is 0.999 with a median 1.0, and standard deviation of 4.5e-07, as expected of an accurate and well formed caption.

### 6.1.3. METRICS

Each agent outputs included a generated image, a final caption and a final belief graph. We evaluate each modality via their alignment to the ground truth image $\mathbf{I}$, caption $c$ and belief $b_{gt}$, using the following metrics.

**Text-Text Similarity**: We use the following metrics to compare the ground truth caption and the merged prompt from the agent: 1) **T2T** – embedding-similarity computed using Gemini 1.5 Pro[4] and 2) **DSG** (Cho et al., 2024) adapted to parse text prompts into Davidsonian scene graph using the released code.

**Image-Image Similarity (I2I)**: We compute cosine similarity between the groundtruth image and the generated image from the agent prompt. We use image features from DINOv2 (Oquab et al., 2024) model following prior works.

**Text-Image Similarity (T2I)**: We compare the ground truth prompt with the generated image using VQAScore (Lin et al., 2024). We use the author released implementation of the metric and use Gemini 1.5 Pro as the underlying multi-modal large language model.

**Negative log likelihood (NLL)**: We construct the ground truth state of the image in the form of a belief graph but with no uncertainty. We then approximately compute the NLL of the ground truth state given the belief of the agent at each turn, by assuming the independence of all entities, attributes and relations, and summing their log probabilities.[5]

### 6.2. Automatic Metrics and Qualitative Results

The automatic evaluations across COCO-captions, ImageIn-Words, DesignBench datasets show similar results and highlight the same patterns across the different agents.

Results in Table 1 show the $\mathbf{I}$, $c$ and $b_{gt}$ against each agent's final generated image, merged prompt and belief graph. All

columns except 'human rating' show the mean and standard deviation of the similarity metric at the final turn. The blue row shows the baseline T2I method which generates an image directly from the *starting prompt* $p_0$.

**Multi-Turn agents show clear advantage:** The results in Table 1 show that significant gains in performance come from using proactive multi-turn agents. The blue row shows the simplest baseline which directly uses a T2I model and performs no updates to the initial prompt $p_0$. We see that all of the multi-turn agents far exceed the baseline T2I model on both datasets and all metrics. Ag3 (the LLM agent that does not explicitly utilize the belief graph to generate questions) shows superior performance across all metrics. This confirms that the current T2I agents often produce less desirable images given ambiguity in prompts. In Figure 2 we see real outputs of the multi-turn set up with the Ag3 agent. Additionally, we see that the multi-turn agents generally improve in most metrics as they increase the number of interactions. Figure 3 and **??** show the I2I, T2I, T2T and NLL metrics, averaged across all images in the ImageIn-Words dataset, per turn for 15 turns. Interestingly we see the T2T and the T2I VQA similarity scores seem to plateau or decrease after about 10 interactions, while the I2I scores continue to increase. The NLL metric shows large performance gains of the Ag3 agent in comparison to all other methods. Figure 11 shows the T2T DSG metrics.

**LLMs play a significant role:** The best performers (Ag2 and Ag3) both query the LLM to generate a question based on contextual information such as the belief graph and conversation history. They query the LLM to construct a concise and clear question but don't impose further constraints on the question construction. Ag1 provides a programmatic template for how the LLM should construct the question based on its belief graph and does not provide any conversation history information. Examples of dialogs and the generated questions produced by the three agents can be found in the Appendix in Figure 5. This figure demonstrates that the templated question creation sometimes leads to questions that gather minimal information in return. This can be an intrinsic limitation of hard coded question selection strategy but also can be an issue of the heuristic scores we defined for question selection in Ag1. In contrast, Ag2 and Ag3 generate questions that are more open-ended thus allowing the user to provide more nuanced details which in consequence enhance the agent's image knowledge.

**Question prompts with question-asking principles show advantage over those with beliefs:** The Ag3 agent (which uses an LLM with question generation instructions about entity, attributes etc related to the belief) dominates across all datasets on almost every metric. Ag2 uses the belief explicitly to construct questions by passing the belief into the LLM as information from which to generate the next

---

[4]Text embeddings are obtained from Embeddings API: https://ai.google.dev/gemini-api/docs/embeddings.

[5]This approximation does not account for potential similarities in the names of entities or attributes. This could lead to approximation errors if, for example, the model confuses "Persian cat" with "Siamese cat" due to their similar names. Addressing this limitation would require incorporating semantic similarity measures into the NLL computation.

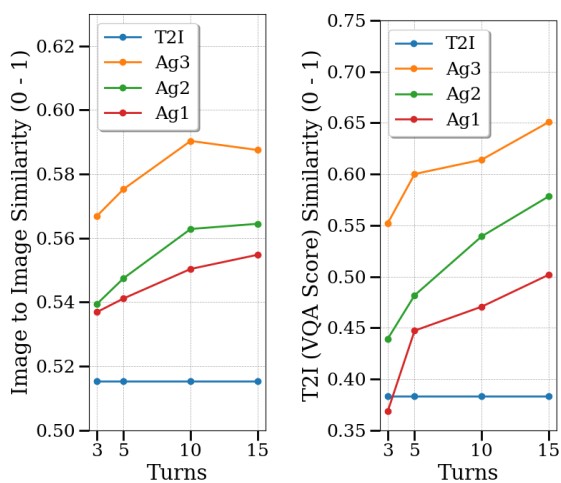

*Figure 3.* I2I and T2I metrics per interaction turn on ImageInWords. Out proactive agents outperform standard setups of T2I.

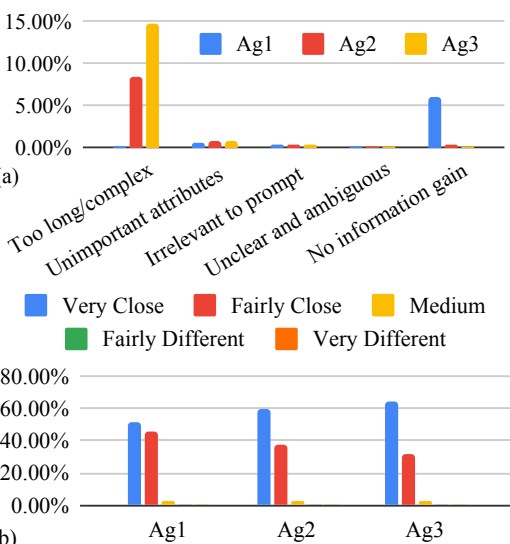

*Figure 4.* Human study results on the dialogues between agents and simulated users. (a) Issues in each agent's questions, as determined by human raters. (b) Ratings of how well the final generated image corresponds to the user prompt and dialogue.

question. When inspecting the reasoning steps of Ag2, we found that Ag2 excessively relies on importance scores in beliefs to ask questions, and if the importance scores are not estimated properly, the quality of the questions decreases.

### 6.3. Human Studies on Generated Images and Dialogues

We conducted human studies to provide more insights that complement the quantitative results from our automatic evaluations. See detailed design in Figure 19, Figure 20 and Figure 21. Information on human user recruitment and consent can be found in §I.

Table 1 shows the human rater evaluations on the generated images in comparison to the ground truth image. Participants are asked to rank the images produced by the three proposed multi-turn agents and a single-turn T2I model in terms of content and aesthetics/style against a ground truth image for which the original prompt was derived and the answers to the agent's questions were derived. The human ratings correspond to the I2I (DINO) metric. Approximately 550 image-dialog pairs per agent are rated using 3 human raters. The generated unlabeled images were presented in a random order and the human rater was tasked with ranking the images from best to worst in terms of content correctness. The results are in Table 1, under the column *Human Rating*, showing that agentic systems are selected as the best generated image over the single-turn T2I in 80%+ of cases for both content and aesthetics/style.

Figure 4 shows the human study results on the generated agent-simulated user dialogues. (1) Per question raters are asked to mark issues each agent question contains that could pose a disturbance to the user, such as 'question is too long and complex'. Approximately 8k questions per agent are rated. The results in Figure 4 (a) show agentic systems have issues with their questions in 14% or less cases. Results

show Ag2 and Ag3 suffer from too long questions while Ag1 suffers from questions that do not gain new information. (2) Raters are asked to evaluate the correspondence of each image to the agent-user dialog and original prompt. Approximately 1.5k image-dialog pairs are rated using 3 human raters. Results in Figure 4 (b) show that for all of our agents, more than 96% of the 1.5k image-dialog pairs are rated as very close or fairly close with some differences. This high rating shows the viability of the T2I model employed by the agents, as well as the agents' ability to combine the dialogue into a coherent prompt to feed the T2I model.

### 6.4. Human Studies on the Agent Interface

We performed a human survey for understanding user frustrations and validation. We gathered data from 143 participants who all identified to be regular T2I users (at least once a month). Participants were presented with four hypothesized frustrations (prompt misinterpretation, many iterations, inconsistent generations, incorrect assumptions) and three potential mitigating features (clarifications, entity graph, relationship graph; more details in §H).

Table 4 in Appendix confirms the prevalence of hypothesized frustrations amongst users, with 83% experiencing occasional, frequent, or very frequent frustration due to prompt iterations, followed by 70% for misinterpretations, 71% for inconsistent generations, and 60% experiencing frustration due to incorrect assumptions. Most acutely 55% of participants reported frequent or very frequent frustration due to the prompt iteration frequency necessary. In Table 2, we report the mitigation features that are likely to help. Clarifications were reported as having the highest likelihood to help current workflows (91% could / likely / very likely to

| Dataset | Model | (EmbedSim) T2T ↑ | (DSG) T2T ↑ | (DINO) I2I ↑ | (VQAScore) T2I ↑ | NLL↓ | (Aesthetics) Human Eval↑ | (Content) Human Eval↑ |
|---|---|---|---|---|---|---|---|---|
| Coco-Captions | T2I | 0.876±.03 | 0.590±.1 | 0.517±.2 | 0.298±.5 | 520.065±161 | 0.103 | 0.184 |
| | Ag1 | 0.944±.02 | 0.756±.1 | 0.627±.1 | 0.583±.5 | 508.401±158 | 0.133 | 0.203 |
| | Ag2 | 0.946±.02 | 0.834±.1 | 0.614±.1 | 0.663±.5 | 481.722±154 | 0.253 | **0.316** |
| | Ag3 | **0.950**±.02 | **0.900**±.1 | **0.658**±.1 | **0.775**±.4 | **446.568**±151 | **0.511** | 0.297 |
| ImageInWords | T2I | 0.881±.02 | 0.681±.1 | 0.515±.2 | 0.371±.5 | 459.905±200 | 0.156 | 0.215 |
| | Ag1 | **0.943**±.02 | 0.816±.1 | 0.555±.2 | 0.5058±.5 | 449.893±196 | 0.201 | 0.235 |
| | Ag2 | 0.938±.02 | 0.879±.1 | 0.565±.2 | 0.570±.5 | 444.523±192 | 0.277 | **0.307** |
| | Ag3 | 0.942±.02 | **0.912**±.1 | **0.588**±.1 | **0.662**±.5 | **429.464**±195 | **0.365** | 0.244 |
| DesignBench | T2I | 0.874±.02 | 0.607±.1 | 0.544±.12 | 0.353±.5 | 320.890±94 | 0.032 | 0.127 |
| | Ag1 | 0.937±.02 | 0.829±.1 | 0.594±.1 | 0.685±.5 | 295.197±69 | 0.175 | 0.190 |
| | Ag2 | 0.938±.02 | 0.918±.1 | 0.642±.1 | 0.855±.3 | 271.260±82 | 0.159 | 0.317 |
| | Ag3 | **0.943**±.02 | **0.949**±.0 | **0.692**±.1 | **0.955**±.2 | **257.435**±68 | **0.635** | **0.365** |

*Table 1.* Automatic evaluation results on **Coco-Captions**, **ImageInWords**, and **DesignBench**. Our agents show large performance gains in all metrics over a standard T2I model. The Human Eval columns display the % of times an image was ranked as closest to the ground truth image in terms of content or aesthetics. All other columns correspond to automatic evaluations using metrics described in 6.1.3.

| Helpfulness Rating | Clarification | Entity | Relation |
|---|---|---|---|
| **V. Likely (%)** | 21.7 | 20.3 | 20.3 |
| **Likely (%)** | 37.8 | 32.9 | 28.7 |
| **Could Help (%)** | 31.5 | 35 | 37.1 |
| **Unlikely (%)** | 5.6 | 7.7 | 7.0 |
| **V. Unlikely (%)** | 3.5 | 4.2 | 7.0 |

*Table 2.* Perceived helpfulness of each proposed feature. Shown as (% of users) out of 143 human raters.

be helpful), followed by entity graphs (88% could / likely / very likely to be helpful) and relationship graphs (86% could / likely / very likely to be helpful). Clarifications were expected to deliver value immediately / very soon by 58%.

These findings highlight a strong user desire for, and a high likelihood of success for, features that make T2I generation more efficient and user-friendly by reducing iterations and mitigating misinterpretations. Full explanations of the hypothesized frustrations, mitigation and responses splits are in §H. All respondents were compensated for their time as per market rates, and were recruited by our vendor to ensure diversity across age, gender, and T2I usage in terms of models, frequency and purpose (work and non work).

## 7. Discussion and Conclusion

This work introduces a design for agents that assist users in generating images through an interactive process of proactive question asking and belief graph refinement. By dynamically updating its understanding of the user's intent, the agent facilitates a collaborative approach to image generation with fewer user frustrations. Moreover, presenting the agent's belief graph can be a generalizable method for AI transparency, which is an important factor given the increasing complexity of modern AI models.

**Modular design.** Our agent prototypes are highly modular: they use frozen T2I models to generate images based on prompts updated by the agents. This allows for seamless integration of improved off-the-shelf T2I models as they become available, boosting system performance without requiring further adaptation.

While T2I prompt-image alignment errors can limit the effectiveness of our proposed agents, our experiment in §F.2 demonstrates that by leveraging agent-user QA pairs to evaluate image fidelity, we can filter out misaligned images across N seeds, thereby mitigating the impact of these errors. Note that the agents' T2T scores in Table 1 (92%+) ablate the T2I model, showing that they have already reached high alignment on the caption level, which is bound to increase with improved T2I capabilities.

**Culturally competent models**: There has been a growing use of generative AI models by a broader section of society, which is only expected to continue to grow. With the varied preferences and intents across diverse user bases, this work is an example of steerable models that can adapt better to diverse user needs. Moreover, different groups of people may perceive harmfulness of content differently, so learning more about the user through clarification questions can mitigate risks, make models safer, and pave the way to more inclusive generative AI systems.

**Future work.** It would be interesting to explore generating images directly from belief graphs and fine-tuning VLMs on image-text interleaved multi-turn trajectory data. This may require a) collecting data such as gold-standard trajectories or annotations on the quality of trajectories of human-agent conversations and b) approaches to fine-tune the model on multi-turn trajectories of images and text, which can potentially improve the performance of the agent.

## Impact Statement

Our proposed T2I agents are equipped with better tools (belief graphs) for interpretability and controllability. Presenting the agent's belief graph can be a generalizable method for AI transparency, which is an important factor given the increasing complexity of modern AI models.

By asking clarification questions, our proposed agents may enable a more customizable and personalized content creation experience. Because different groups of people may perceive harmfulness of contents differently, learning more about the user through clarification questions can potentially mitigate risks of generating contents that can be offensive to each specific user.

## Acknowledgements

We would like to thank Jason Baldridge and Zoubin Ghahramani for insightful discussions on multi-turn T2I and belief states, Mahima Pushkarna for the help and consultation on user study. We would also like to thank Richard Song, Noah Fiedel, Susanna Ricco and anonymous reviewers for feedback on the paper.

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

## A. Author contributions

All authors contributed to writing.

- Meera Hahn (meerahahn@google.com): Led automated evaluation and human evaluation of agents. Created DesignBench. Contributed to agent design and development.
- Wenjun Zeng (wenjunzeng@google.com): Significant contribution to agent design and development, as well as open-sourcing.
- Nithish Kannen (nitkan@google.com): Significant contribution to experiments and early versions of agents. Led open-sourcing.
- Rich Galt (richgalt@google.com): Led agent interface design and human studies.
- Kartikeya Badola (kbadola@google.com): Contributed to experiments and early versions of agents.
- Been Kim (beenkim@google.com): Advised project direction with critical feedback.
- Zi Wang (wangzi@google.com): Proposed and initiated project. Led agent design and development. Advised project. Contributed to evaluation.

## B. Related Work

From the very outset of **artificial intelligence**, a core challenge has been to develop intelligent agents capable of representing knowledge and taking actions to acquire knowledge necessary for achieving their goals (McCarthy & Hayes, 1969; Minsky, 1974; Moore, 1985; Nilsson, 2009; Russell & Norvig, 2016). Our work is an attempt to address this challenge for intelligent T2I agents.

In **machine learning and statistics**, efficient data acquisition has been extensively studied for many problems, including active learning (Cohn et al., 1996; Settles, 2009; Houlsby et al., 2011; Gal et al., 2017; Ren et al., 2021; Wang et al., 2018), Bayesian optimization (Garnett, 2023; Kushner, 1964; Močkus, 1974; Auer, 2002; Srinivas et al., 2010; Hennig & Schuler, 2012; Wang & Jegelka, 2017; Wang et al., 2024b), reinforcement learning (Kaelbling et al., 1996; Ghavamzadeh et al., 2015; Sutton, 2018) and experimental design (Chaloner & Verdinelli, 1995; Kirk, 2009). We reckon that T2I agents should also be capable of actively seeking important information from human users to quickly reduce uncertainty (Wang et al., 2024c) and generate satisfying images. In §E, we detail the implementation of action selection strategies for our T2I agents.

In **human-computer interaction**, researchers have been extensively studying how to best enable Human-AI interaction especially from user experience perspectives (Norman, 1994; Höök, 2000; Amershi et al., 2019; Cai et al., 2019; Viégas & Wattenberg, 2023; Chen et al., 2024; Yang et al., 2020; Kim et al., 2023). Interface design for AI is becoming increasingly challenging due to the lack of transparency (Viégas & Wattenberg, 2023; Chen et al., 2024), uncertainty about AI capability and complex outputs (Yang et al., 2020). We aim to build user-friendly agents, and an indispensable component is their interface to enable them to effectively act and observe, as detailed in §G.

**Interpretability.** Surfacing an agent's belief overlaps with interpretability as both aim to understand model or agent's internal state. Some methods leverage LLM's natural language interface to surface their reasoning (e.g., chain of thought (Wei et al., 2023a)), sometime interactively (Wang et al., 2024a). While these approaches make accessible explanations, whether the explanations represent the truth has been questioned (Lanham et al., 2023; Wei et al., 2023b; Chen et al., 2023). Some studies indicate explanations generated by the LLMs may not entail the models' predictions nor be factually grounded in the input, even on simple tasks with extractive explanations (Ye & Durrett, 2022). In this work, the belief graph does not correspond to the distribution over outputs of the T2I *model* itself conditioned on the underspecified prompt. Instead, the belief graph is designed to align with the distribution over image-prompts generated by the *agent*, since the agent can construct detailed prompts according to its belief, and feed them into a high-quality T2I model.

**Text-to-Image (T2I) generation.** Text-to-image prompts can be ambiguous, subjective (Hutchinson et al., 2022), or challenging to represent visually (Wiles et al., 2024). Different users often have distinct requirements for image generation, including personal preferences (Wei et al., 2024), style constraints (Wang et al., 2023), and individual interpretations (Yin et al., 2019). To create images that better align with users' specific needs and interpretations, it is essential to actively communicate and interact with the user to understand the user's intent. Other work (Mehrabi et al., 2022) proposes to use clarification questions to resolve ambiguities (a prompt has multiple meanings), but our aim is to use clarification questions to resolve under-specification (the prompt is not ambiguous, but lacks information to fully describe the image).

T2I models are evaluated by analyzing **image-prompt alignment** using metrics that can be embedding-based, such as

CLIPScore (Hessel et al., 2022), ALIGNScore (Zha et al., 2023), VQA-based such as TIFA (Hu et al., 2023), DSG (Cho et al., 2023) and VQAScore (Lin et al., 2024), and captioning-based like LLMScore (Lu et al., 2023). Approaches such as PickScore (Kirstain et al., 2023), ImageReward (Xu et al., 2023) and HPS-v2 (Wu et al., 2023) finetune models on human ratings to devise a metric that aligns with human preferences. Diversity of generated images (Naeem et al., 2020) is also used as a metric to track progress, especially in the geo-cultural context (Kannen et al., 2024; Hall et al., 2024). To evaluate our agent-user conversations and corresponding generated images, we develop an automatic approach using a variety of the metrics mentioned as well as collect human ratings on the generated images and generated conversations.

**Prompt expansion** is a widely known technique to improve image generation (Betker et al., 2023). ImageinWords (Garg et al., 2024) proposes to obtain high-quality hyper-detailed captions for images, which significantly improve the quality of image generation. Datta et al. (2024) present a generic prompt expansion framework used alongside Text-to-Image generation and show an increase in user satisfaction through human study. While our work can be viewed as a method to adaptively expand a T2I prompt based on user feedback[6], evaluating our method as a prompt expansion tool is outside of our scope.

## C. Formalism of the Agent and its Objective

We define an interactive agent as a $\langle B, A, O, \tau, \pi \rangle$ tuple, where we have

- $S$: a state space where each state represents an intent,
- $B$: a space of agent belief states,
- $A$: a space of actions that the agent can take,
- $O$: a space of agent observations of the user,
- transition function $\tau : B \times A \times O \mapsto B$ for updating beliefs given new interactions,
- action selection strategy $\pi : B \mapsto A$, which specifies which action to take given a belief.

For each user-initiated interaction, we assume that there exists a specific intent $s \in S$ in hindsight, where $S$ is the space of all possible user intents. For a T2I task, this intent is the image the user would like to generate at the end of the interaction.

Each type of T2I agents can have a unique user intent representation, belief representation, construction of the action space, and user interface design to obtain observations of users.

In §4, we show the examples for these components.

We use a score function, $f : B \times S \mapsto \mathbb{R}$, to evaluate the alignment between an agent belief and a user intent at any turn of the interaction. Function $f$ can only be evaluated in hindsight once the user intent is revealed. The agent does not have direct access to function $f$ since the user intent is hidden from the agent. However, the agent may construct a probabilistic distribution over function $f$ based on its belief about the user intent. The goal of the agent is to maximize function $f$ with as few turns of interaction with the user as possible.

## D. Visualization of Multi-Turn Agent-User Dialogs and Generated Images

In Figure 5, we show examples of multi-turn dialogs between simulated users and the three agents in Section 6. We also visualize the generated images in Figure 6, Figure 7, Figure 8 and Figure 9.

## E. Implementation Details of Agent Prototypes

We propose three T2I agents, each characterized by a unique configuration of $\langle B, A, O, \tau, \pi \rangle$ tuples:

- *Ag1: Heuristic Score Agent*. This agent incorporates a human-defined heuristic score based on the belief to guide question generation. This heuristic score reflects the perceived importance of different aspects of the belief in driving the conversation forward;
- *Ag2: Belief-prompted Agent*. This agent leverages an LLM to generate questions by processing both the conversation history and a structured representation of the belief.
- *Ag3: Principle-prompted Agent*. This agent generates questions directly from the conversation history based on the principles introduced in §4.2. The question asking strategy of Ag3 relies solely on the implicit knowledge and reasoning

---

[6]Samples from the agent belief can be used to construct expanded prompts.

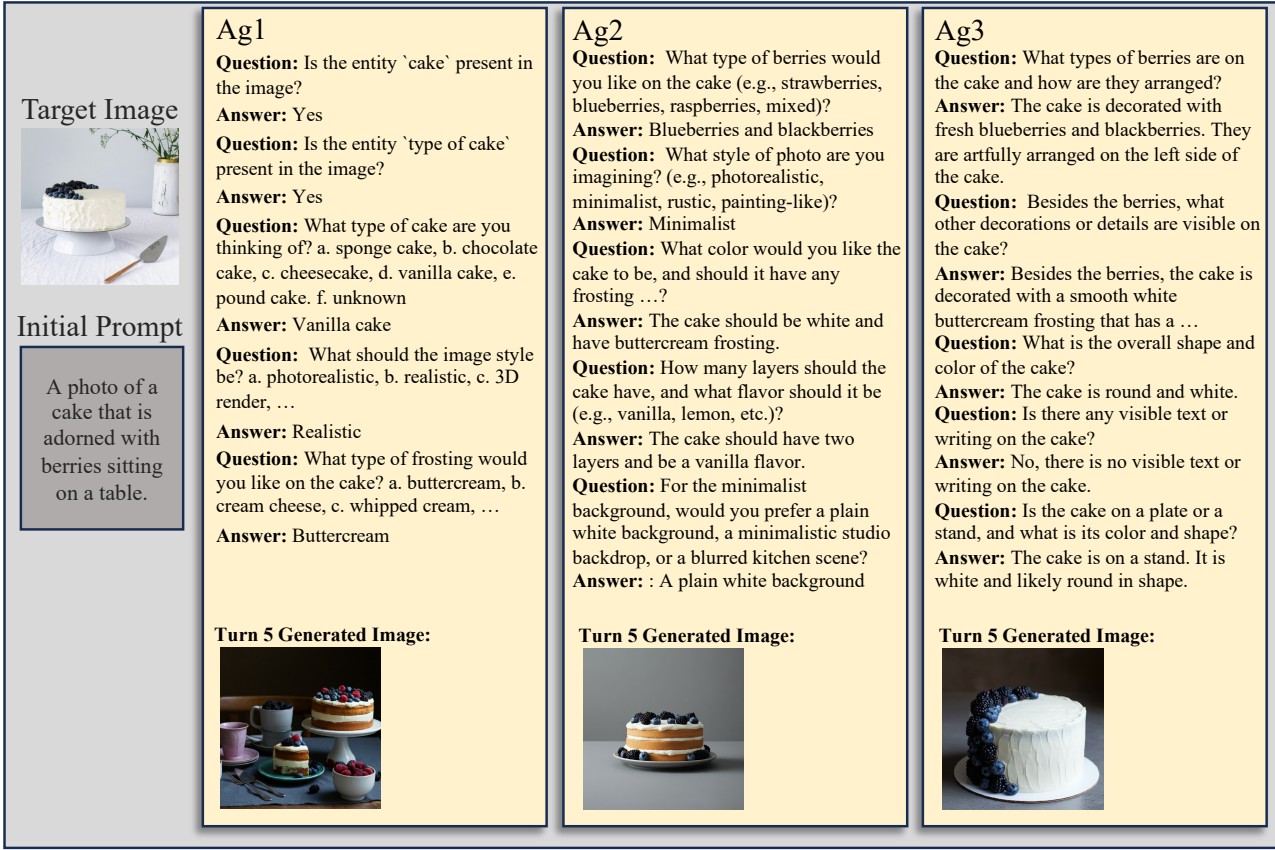

*Figure 5.* Real multi-turn dialogs generated by the Ag1, Ag2, and Ag3 agents on an image from DesignBench. The figure additionally shows the image generated after the 5 turn dialog per agent.

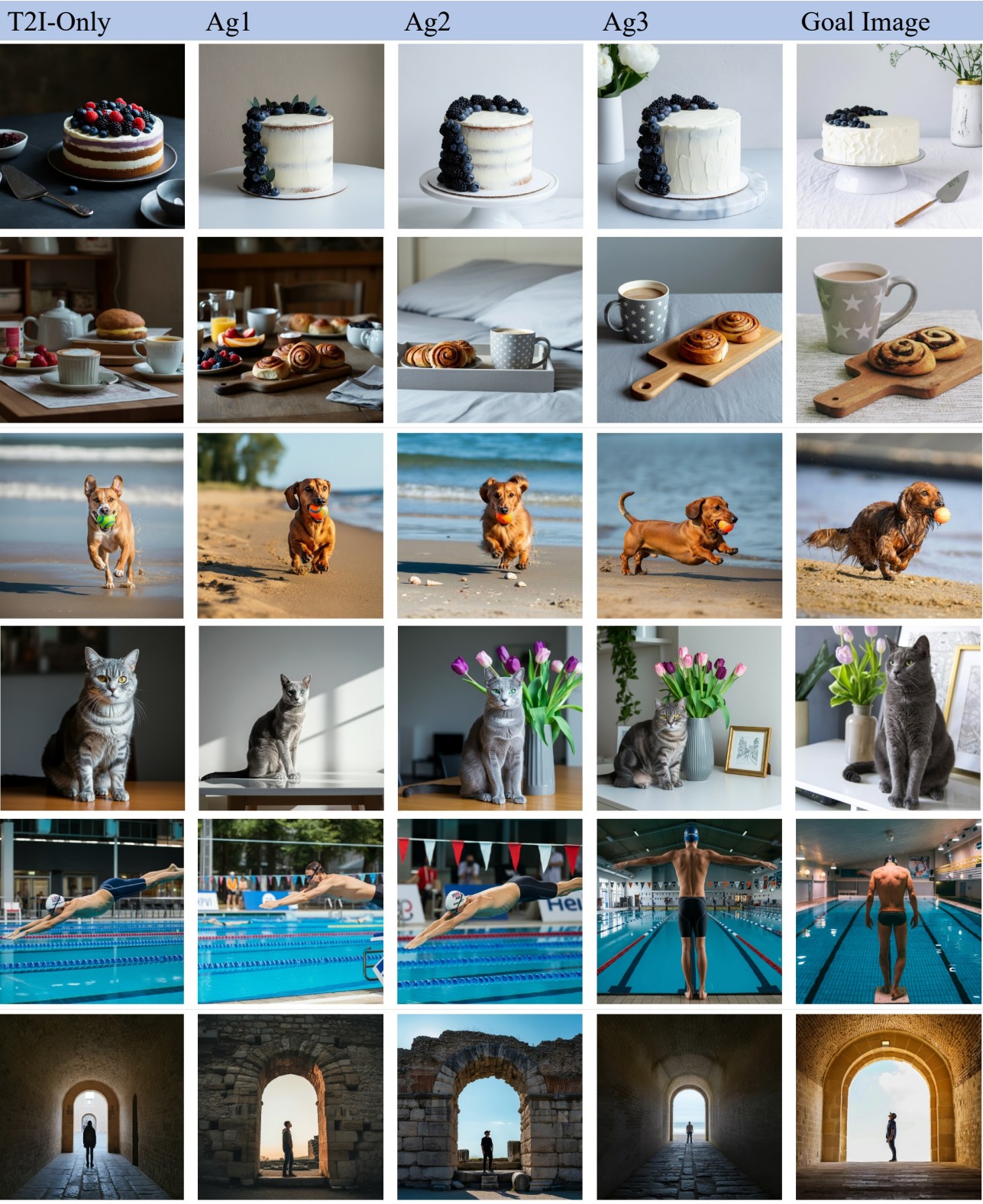

*Figure 6.* Agent Generated Image Outputs on DesignBench: a chart of the generated image outputs of the four main Agent types in comparison to the goal image. Each column displays the output of a different agent and the right most column shows the goal image that the agents aimed to recreate. Each agent was provided with the same starting prompt and iterated for 15 turns, with the exception of the "T2I" agent column which produces an image from the starting prompt. Ag1, Ag2 and Ag3 refer to the Agents described in §E. Each agent uses the same T2I model to produce the final image. The goal images displayed here are from our DesignBench dataset described in the experiments section.

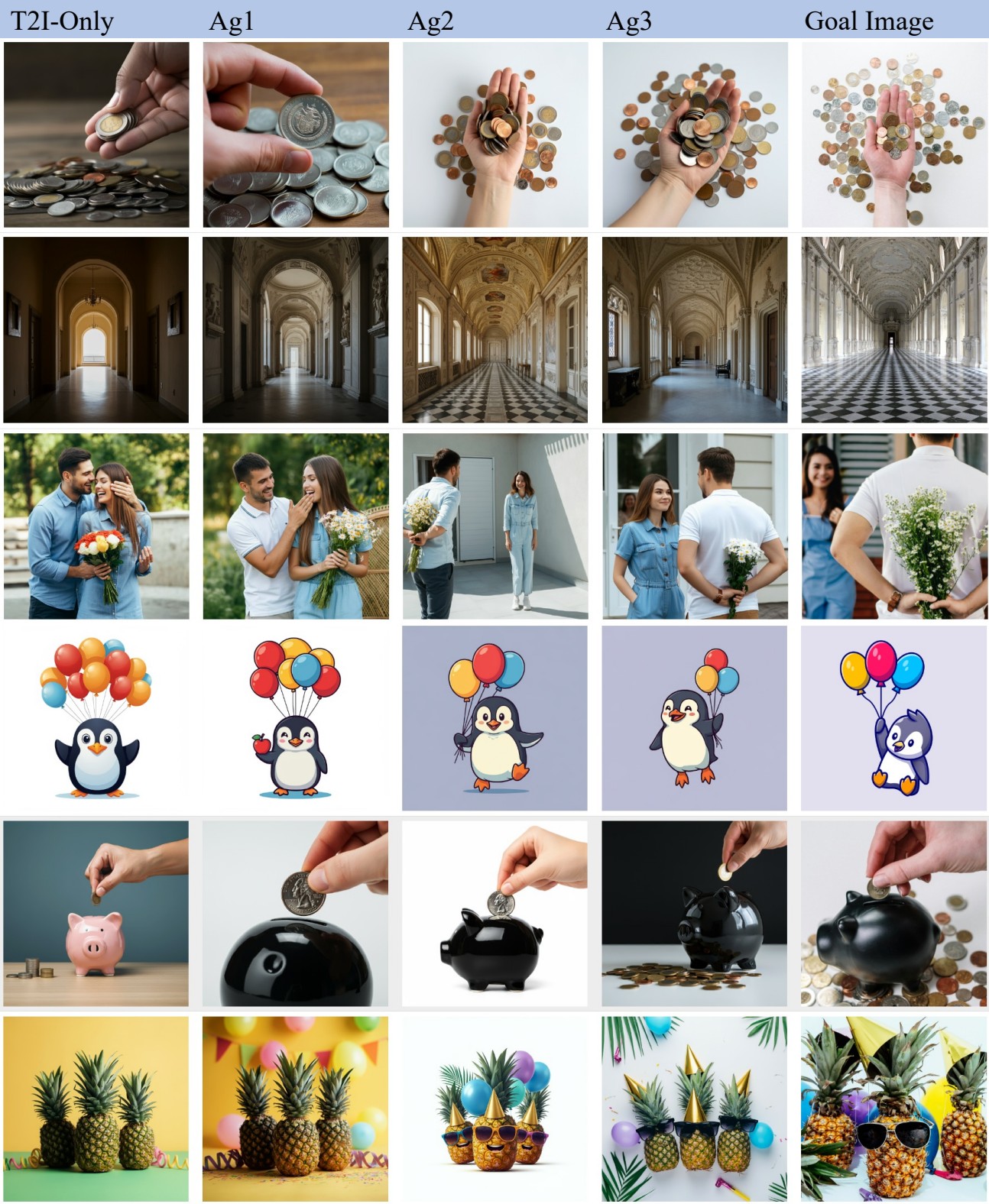

*Figure 7.* Agent Generated Image Outputs on DesignBench (Continued): a chart of the generated image outputs of the four main Agent types in comparison to the goal image. Each column displays the output of a different agent and the right most column shows the goal image that the agents aimed to recreate. Each agent was provided with the same starting prompt and iterated for 15 turns, with the exception of the "T2I" agent column which produces an image from the starting prompt. Ag1, Ag2 and Ag3 refer to the Agents described in §E. Each agent uses the same T2I model to produce the final image. The goal images displayed here are from the DesignBench dataset described in the experiments section.

| T2I-Only | Ag1 | Ag2 | Ag3 | Goal Image |
| --- | --- | --- | --- | --- |

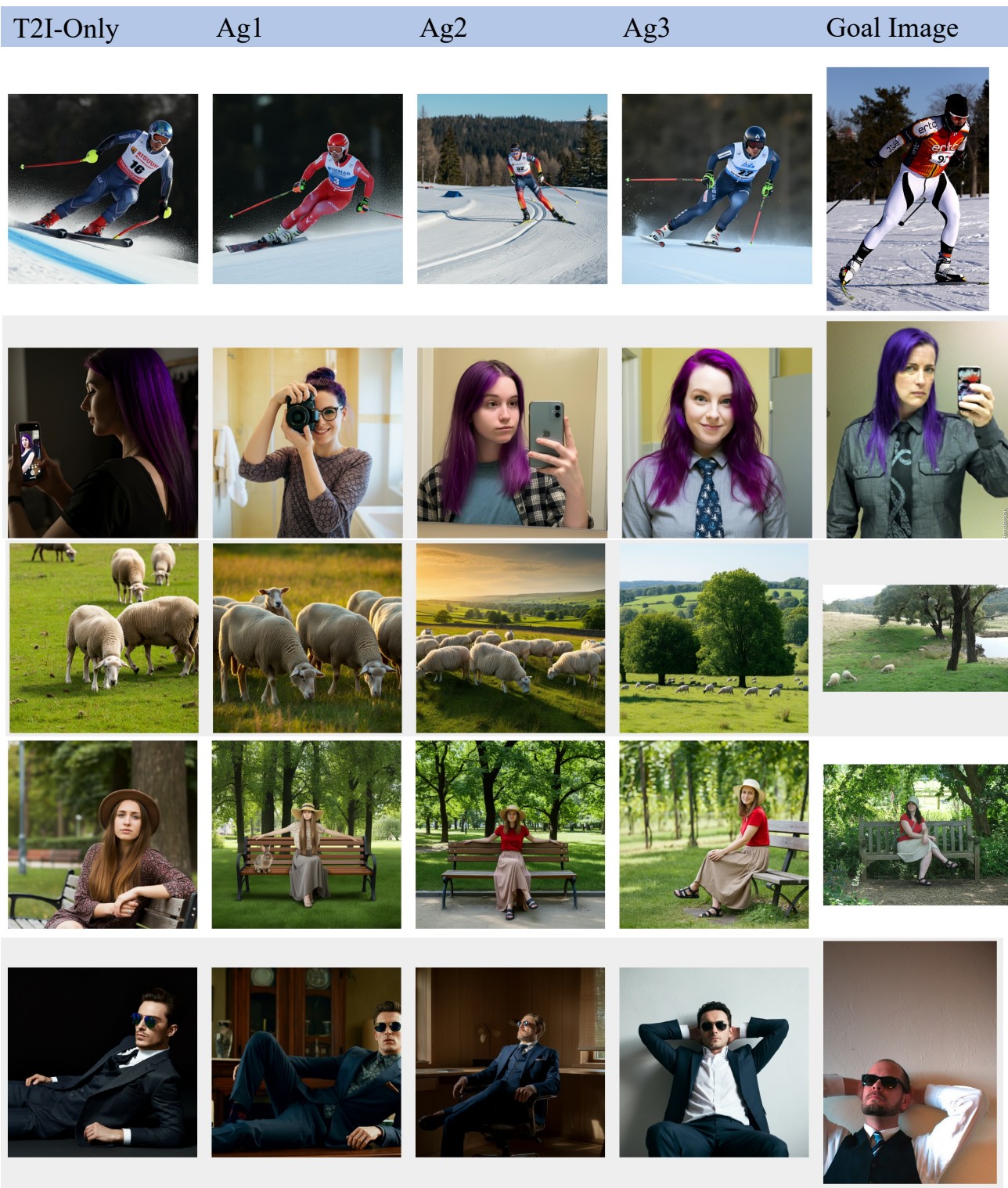

*Figure 8.* Agent Generated Image Outputs (Coco-Captions Validation): a chart of the generated image outputs of the four main Agent types in comparison to the goal image. Each column displays the output of a different agent and the right most column shows the goal image that the agents aimed to recreate. Each agent was provided with the same starting prompt and iterated for 15 turns, with the exception of the "T2I" agent column which produces an image from the starting prompt. Ag1, Ag2 and Ag3 refer to the Agents described in §E. Each agent uses the same T2I model to produce the final image. The goal images displayed here are from the Coco-Captions (Chen et al., 2015) dataset described in the experiments section.

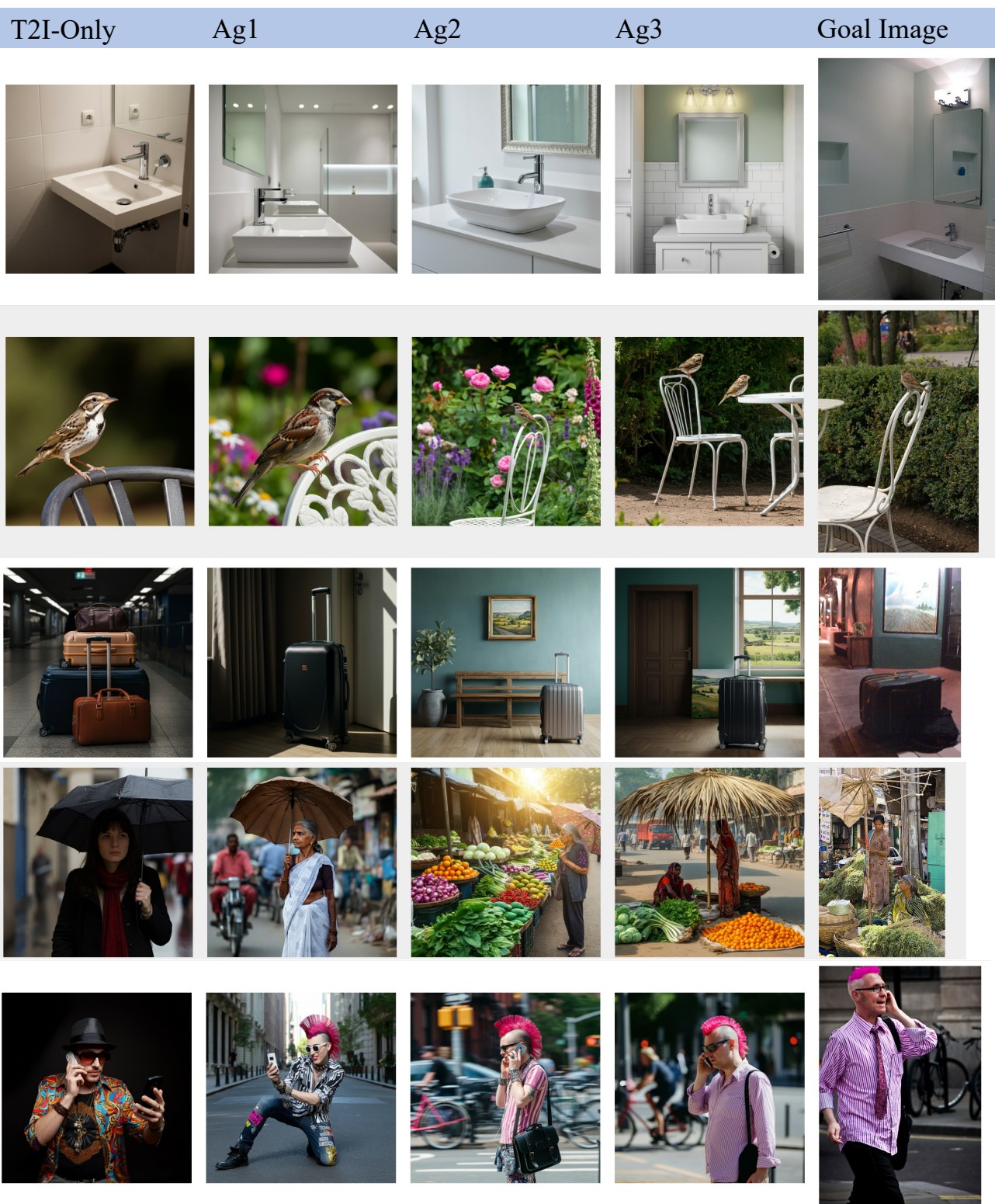

*Figure 9.* Agent Generated Image Outputs (Coco-Captions Validation): a chart of the generated image outputs of the four main Agent types in comparison to the goal image. Each column displays the output of a different agent and the right most column shows the goal image that the agents aimed to recreate. Each agent was provided with the same starting prompt and iterated for 15 turns, with the exception of the "T2I" agent column which produces an image from the starting prompt. Ag1, Ag2 and Ag3 refer to the Agent described in the methods section. Each agent uses the same T2I model to produce the final image. The goal images displayed here are from the Coco-Captions (Chen et al., 2015) dataset described in the experiments section.

capabilities of the underlying LLM.

A summary of each agent is located in §5.

### E.1. Implementation of Agent Beliefs

Technically, an agent belief $b \in B$ is represented in two complementary forms: (i) Merged prompt: This is a natural language representation that summarizes the entire conversation history up to the current turn. It provides a comprehensive textual overview of the user's requests, feedback, and any clarifications exchanged with the agent. (ii) Belief graph: This is a symbolic representation derived from the merged prompt. It parses the natural language text into a structured format, capturing key elements like entities, attributes, relationships, and associated probabilities. This structured representation facilitates more precise reasoning and decision-making by the agent.

**Prompt Merging.** An LLM (§E.8) summarizes the latest interaction, encapsulating the agent's question and the user's response into a concise textual representation. This step distills the essential information exchanged during the interaction. Another LLM (§E.9) merges the existing merged prompt (containing the accumulated information from previous interactions) and the summarized interaction at the current turn. This creates an updated prompt that reflects the evolving understanding of the user's intent.

**Belief Parsing.** See an example of the belief graph in Figure 10. We employ three specialized parsers trained via in-context learning (ICL): entity parser (§E.5) analyzes the user prompt to identify and extract a list of relevant entities.; attribute parser (§E.6) takes user prompt and an entity as the input to extract a list of attributes associated with that entity; relation parser (§E.7) takes the user prompt and a list of entities as input and identifies relationships between those entities. Each entity is associated with meta information like name, importance to ask score, description, probability of appearing, a list of attributes like color, position, etc [7]. Each attribute contains meta information like name, importance to ask score, a list of possible values for the attribute along with their associated probabilities, etc. Each relation includes meta information such as: name, description, spatial relation, importance to ask score, entity 1 and entity 2, whether the relation is bidirectional, etc.

### E.2. Implementation of Action Selection

From an information theoretic perspective, an optimal action is the one that maximizes the information gain between the observation and the belief, i.e. $a_t = \arg\max_a H(o_{i-1}; b_{i-1} \mid a) - H(o_i; b_i \mid a)$. However, directly optimizing this objective can be computationally challenging. Therefore, we explore several heuristic strategies to effectively reduce uncertainty:

- Maximize the overall heuristic importance score ($MHIS$): This strategy focuses on maximizing the overall importance score of the entities, attributes, and relations within the belief. We further ask a question regarding an attribute or relation by maximizing the overall heuristic importance score. The score can be modeled as:

$$max_{e,a,c,r}(IS(e) * IS(a) * P(e) * Ent(c), IS(r) * P(r) * Ent(c)) \tag{1}$$

  Here $IS, P, Ent$ represent importance to ask score, probability of appearing, and entropy of the probabilities respectively and $e, a, c, r$ represent entity, attribute, candidate list, relation respectively.
- Ask Important Clarification Question based on belief ($AICQ_B$): This strategy leverages the structured information within the belief. We provide the LLM with the user prompt, conversation history, and the current belief, utilizing an ICL prompt (§E.11) to guide question generation. The LLM then formulates a clarification question aimed at eliciting information about key features of the image, naturally prioritizing those with higher *Importance to ask score* within the belief.
- Ask Important Clarification Question directly ($AICQ_{base}$): This strategy relies on the LLM's inherent ability to identify important aspects of the user prompt and conversation history. The LLM (§E.10) generates an important clarification question based on its implicit understanding of the user's needs, without explicitly relying on the structured information in the belief.

*Ag1* employs $MHIS$ strategy for question generation. This strategy leverages the importance scores assigned to entities, attributes, and relations within the belief graph. It identifies the element with the highest heuristic importance score and

---

[7]**Name** is a unique identifier for the entity; **Importance to ask score**: A numerical value indicating the entity's perceived importance in satisfying the user's request. Entities with higher scores are prioritized during question generation, as they are likely to reduce uncertainty and contribute significantly to the final image; **Description** provides a textual description of the entity; **probability of appearing** estimates likelihood of the entity being present in the generated image; **Attributes** are for understanding the detailed attributes of the entities.

Entities

Rabbit
Attribute Name: color, importance_score: 0.9,
        candidates: [brown: 0.25, white: 0.25, grey: 0.2, black: 0.15, …]
Attribute Name: breed, importance_score: 0.3,
        candidates: [Dutch: 0.2, Mini Lop: 0.15, Netherland Dwarf: 0.15, …]
Attribute Name: expression, importance_score: 0.5,
        candidates: [scared: 0.8, determined: 0.1, playful: 0.1]
                        ……

Dog
Attribute Name: breed, importance_score: 0.8,
        candidates: [Labrador Retriever: 0.15, Golden Retriever: 0.15, German
        Shepherd: 0.15, Bulldog: 0.1, Beagle: 0.1, …]
Attribute Name: coat_color, importance_score: 0.7,
        candidates: [brown: 0.2, black: 0.2, white: 0.2, …]
Attribute Name: coat_style, importance_score: 0.6,
        candidates: [long: 0.3, short: 0.3, fluffy: 0.2, shaggy: 0.1, wavy: 0.1]
Attribute Name: hat, importance_score: 0.5,
        candidates: [baseball cap: 0.2, bowler hat: 0.2, top hat: 0.2, …]
                        ……
                    ……

Relations

Dog-Rabbit
importance_score: 0.9,
spatial_relation: [chasing: 1.0]

Coat-Dog
importance_score: 0.8,
adornment_relation: [wearing: 1.0]
                    ……

*Figure 10.* An example of the belief graph data structure for a given prompt in Figure 1.

formulates a question aimed at eliciting further information about that specific element. The question is then verbalized using the LLM described in Section §E.12.

*Ag2* utilizes the parsed belief graph as the basis for question generation. It employs the $AICQ_B$ strategy, which leverages the structured information within the belief graph to generate targeted clarification questions.

*Ag3* relies solely on the conversation history for question generation. It employs the $AICQ_{base}$ strategy, which leverages the LLM's ability to understand the ongoing dialogue and identify key areas requiring further clarification.

### E.3. User Simulation

To simulate end-to-end agent-user interactions, we implement a user simulator that mimics human question-answering behavior. This simulator operates as follows:

- It generates a belief graph based on a ground truth prompt, representing the user's intended image. This serves as the simulator's internal representation of the desired image.
- The simulator takes the ground truth prompt, conversation history, and its current belief graph as input. It then leverages an ICL prompt (see §E.11) to generate a response to the agent's question. This ensures that the simulator's answers are consistent with its internal belief graph and the ongoing conversation.

### E.4. Implementation of Belief Transition

Both *Ag1* and *Ag2* require belief updating to incorporate new information gained during the interaction in order to compose clarification questions. At each turn, we perform prompt merging to create a comprehensive prompt that summarizes the conversation history. This merged prompt is then used for belief parsing to obtain an updated belief graph. For *Ag2* (and *Ag3*), this updated belief graph directly informs the subsequent interaction. For *Ag1*, it incorporates additional post-processing mechanisms to enhance memory and prevent redundant questioning: (i) Redundancy elimination: If an attribute or relation has already been addressed in the conversation history, the corresponding user response is assigned as the sole candidate with a probability of 1.0, and its importance score is set to 0. This prevents the agent from repeatedly asking about the same information. (ii) Information retention: If an attribute or relation from the conversation history is absent in the updated belief graph, it is explicitly added. This ensures that the agent retains crucial information even if it's not explicitly present in the latest parsed belief graph.

### E.5. Entity Parser Prompt Instruction

```
1   Given a text−to−image prompt list out all the entities that are mentioned in the prompt.
2
3   ∗∗Explicit Entities :∗∗ List all clearly stated entities within the prompt (people, objects, animals, locations, etc.).
4   ∗∗Implicit Entities :∗∗ Identify potential entities that are implied or strongly suggested by the prompt, even if not explicitly mentioned.
5   ∗∗Background Entities :∗∗ Deduce relevant background elements which could impact the image generation from the prompt or context, including :
6       ∗∗Weather:∗∗ If the scene or mood suggests specific weather conditions (sunny, rainy, stormy, etc.).
7       ∗∗Location:∗∗ If a general or specific setting is hinted at (indoors, outdoors, a particular city/landscape, etc.).
8       ∗∗Time of Day:∗∗ If the prompt implies a certain time (dawn, midday, dusk, night).
9       ∗∗Mood or Atmosphere:∗∗ If the prompt evokes a particular emotion or ambiance (joyful, mysterious, peaceful, etc.).
10
11
12  The output should be list and each entry should be formated as a JSON dict with the following fields :
13
14  "name": The name of the entity.
15  "importance_to_ask_score": The importance score of asking a question about this entity to reduce the uncertainty of what the image is given the user prompt. Make sure that this is
            a number between 0 and 1, higher means more important. Consider these factors when assigning scores : 1. Increate the score for entities that are the primary focus or subject
            of the prompt; 2. increase the score for entities that could strongly influence the layout of the image, such as the position or portrayal of other entities in the scene;
            3. significantlydescrease the score for entities that are already well specified in the prompt; 4. significantlyincrease the score for implicit entities that are likely to
            appear in the image and their appearance can significantly impact the image.
16  "description": A short description of the entity.
17  "entity_type": The type of this entitiy. It could be either explicit, implicit, background. No other value is allowed.
18  "probability_of_appearing": The probability of the entity appearing in the image. This is a number between 0 and 1. You should assign a probability with the following rules in mind
            :
19      1. If the prompt says an entity does not exist, assign a 0.0 probability. Because the entity does not exist, you should also assign 0 to importance_to_ask_score of this entity.
20      2. If the prompt indicates an entity definitely exists in the image, assign a 1.0 probability.
21      3. If the prompt does not say anything about the existence of the entity, assign a probability between 0 and 1. This probability is higher if the entity is more likely to appear
            in the image given the context specified by the prompt.
22      4. If the prompt says an entity exists but there is an indication that the entity is not likely to appear in the image, assign a probability between 0 and 1, higher if the entity
            is more likely to appear in the image.
23
24  Below is an example input and output pair :
25  Example1:
26  Input : {{
27      "user_prompt": "generate an image of a lionhead rabbit running on grass with sun shining. There is no trees in the background."
28  }}
29  Output: [
30      {{
```

```
31        "name": "rabbit ",
32        " importance_to_ask_score ": 0.5,
33        " description ": "a lionhead  rabbit ",
34        " entity_type ": " explicit ",
35        " probability_of_appearing ": 1.0
36    }},
37    {{
38        "name": "grass ",
39        " importance_to_ask_score ": 0.5,
40        " description ": "grass ",
41        " entity_type ": " explicit ",
42        " probability_of_appearing ": 1.0
43    }},
44    {{
45        "name": "sun",
46        " importance_to_ask_score ": 0.1,
47        " description ": "sun  is  shining ",
48        " entity_type ": " explicit ",
49        " probability_of_appearing ": 0.3
50    }},
51    {{
52        "name": "sun  light ",
53        " importance_to_ask_score ": 0.1,
54        " description ": "sun  light  shining on the grass  and the  rabbit ",
55        " entity_type ": " explicit ",
56        " probability_of_appearing ": 1.0
57    }},
58    {{
59        "name": " tree ",
60        " importance_to_ask_score ": 0,
61        " description ": " trees  in  the  background",
62        " entity_type ": " explicit ",
63        " probability_of_appearing ": 0
64    }}
65    {{
66        "name": "camera angle ",
67        " importance_to_ask_score ": 0.8,
68        " description ": "the camera angle  of  the  image",
69        " entity_type ": "background",
70        " probability_of_appearing ": 1.0
71    }},
72    {{
73        "name": "weather",
74        " importance_to_ask_score ": 0.8,
75        " description ": "weather",
76        " entity_type ": "background",
77        " probability_of_appearing ": 1.0
78    }},
79    {{
80        "name": "image style ",
81        " importance_to_ask_score ": 1.0,
82        " description ": "the  style  of the image",
83        " entity_type ": "background",
84        " probability_of_appearing ": 1.0
85    }},
86    {{
87        "name": "background color ",
88        " importance_to_ask_score ": 0.8,
89        " description ": "the background color of the image",
90        " entity_type ": "background",
91        " probability_of_appearing ": 0.5
92    }}
93  ]
94
95  ...  [[a few  additional  examples]]  ...
96
97
98  Identify  the  entities  given  the  input  given below.  Strictly  stick  to  the format .
99  Input : {{
100     "user_prompt": "{user_prompt}"
101  }}
102  Output:
```

## E.6. Attribute Parser Prompt Instruction

```
1   Given a text−to−image prompt and a particular  entity  described  in the prompt, and your goal is to  identify  a list  possible  attributes  that could  describe  the  particular  entity .
            Output Requirements:
2
3   1. if  this  attribute  has  already  existed as an entity  in  other  existing  entity  list , then do not include it .
4   2. the  attribute  candidate  could be a mixed of values  like ' color A and color B'.
5   3. The output should be a json parse−able format:
6
7   name ( str ): The name of the  attribute .
8   importance_to_ask_score  ( float ): The importance  score  of asking  a question about this  attribute  to reduce the  uncertainty  of what the image is given the user prompt. This is a
            number between 0 and 1, higher means more  important. Consider these  factors  when assigning scores : 1. Increate  the  score  for  attributes  that are the primary attributes  of an
            important entity ; 2. significantly  increase  the score  for  attributes  that could strongly influence  the  generation  or  portrayal  of OTHER attributes  in  the  scene; 3.
            decrease  the score  for  attributes  that are already well  specified  in the prompt. For example, a breed of a dog would impact other  attributes  like color , size , etc . So the
            breed  attribute  should have a higher importance score than color , size , etc . Assign a much lower score if  the  attribute 's value is already mentioned in the user prompt.
9   candidates ( List of names and probabilities ): List of possible  values  that the  attribute  can take. Make sure to generate  atleast  5 or more possible values . These  should be
            realistic  for  the given entity . For each attribute , returns  the  probability  that the user wants this  candidate based on the user prompt. If  it 's already mentioned by the
```

```
      user , only generate one candidate ( the mentioned one) and assign 1.0 as the  probability . The sum of  probabilities  over all candidates  shall be 1. Also infer the  probability
      based on the prompt. For example, for a dog with breed Samoyed, the color  attribute  has a very high  probability  of white .
10
11  Below are two examples of input and output  pairs :
12
13  Example 1:
14  Input : {{
15    "user_prompt": "generate an image of a white  rabbit  running on grass ",
16    " entity ": " rabbit ",
17    " other_existing_entities  ": "grass"
18  }}
19  Output: [
20      {{
21        "name": "color ",
22        " importance_to_ask_score ": 0.9,
23        "candidates ": {{"white":1.0}}
24      }},
25      {{
26        "name": "breed ",
27        " importance_to_ask_score ": 1.0,
28        "candidates ": {{"Dutch": 0.20,
29                        "Mini Lop": 0.15,
30                        "Netherland  Dwarf": 0.15,
31                        "Lionhead ": 0.10,
32                        "Flemish Giant": 0.10,
33                        "Mini Rex": 0.10,
34                        "English  Angora": 0.08,
35                        "Mini Satin ": 0.05,
36                        "Himalayan": 0.05,
37                        " Californian ": 0.02}}
38      }},
39      {{
40        "name": "age",
41        " importance_to_ask_score ": 0.1,
42        "candidates ": {{"adult": 0.6,
43                        "baby": 0.2,
44                        " senior ": 0.2}}
45      }}
46    ]
47
48  ...  [[a few additional  examples ]]  ...
49
50  Generate  attributes  given the input given below. Do not include other  entities  in the  attributes .  Strictly  stick  to the  format .
51  Input : {{
52    "user_prompt": "{user_prompt}",
53    " entity ": "{ entity .name}",
54    " other_existing_entities  ": "{ existing_entities  }"
55  }}
56  Output:
```

## E.7. Relation Parse Prompt Instruction

```
1   Given a  text−to−image  prompt and a  list  of entity  described in the prompt, your goal is to  identify  a  list  of entity  pairs  that have  relations  between them. Ignore  entity  pairs
       without  relations . The output should be a json  parse−able  format (No comma after the  last  element of the  list ):
2
3   Input :
4   user_prompt: the prompt from the user .
5    entities  :  a  list  of  entities  mentioned in the user_prompt.
6
7   Output:
8   name ( str ): The name of the  relation .  Use ' entity1 −entity2 ' as the  format .
9    description  ( str ): A short  description  of the  relation .
10   spatial_relation  (map from potential  relation  candidates to probability ):  Possible  spatial  relations  between the two  entities .  If  a  relation  is  mentioned in the user prompt, assign
       1.0 as the  probability . The sum of  probabilities  over all  relation  candidates  shall be 1.
11   importance_to_ask_score  ( float ): The importance score of asking a question  regarding this  relation  to reduce entropy. This is a number between 0 and 1, higher means more important.
       Assign a higher score if the two  entities  are very important ,  the  relation  between them is very unclear ,  and the  relation  is very important  for the  layout  of the  image.
12   name_entity_1  ( str ): The name of the  first  entity .
13   name_entity_2  ( str ): The name of the second  entity .
14    is_bidirectional   (bool): Whether the  relation  is  bidirectional .
15
16  Below is an example input and output  pair :
17  Example1:
18  Input : {{
19    "user_prompt": "generate an image of a lionhead rabbit  sitting  on grass , and a eagle is  flying  through the sky. There is a tree in the background.",
20    " entity ": [" rabbit ", "grass ", " eagle ", " tree "]
21  }}
22  Output: [
23      {{
24        "name": " rabbit −grass ",
25        " description ": " rabbit   sitting  on grass ",
26        " spatial_relation  ": {{"above": 0.8, "below": 0.0, "in front of": 0.0, "behind": 0.0, " left of": 0.1, " right  of": 0.1}},
27        " importance_to_ask_score ": 0.1,
28        "name_entity_1 ":" rabbit ",
29        "name_entity_2 ": "grass ",
30        " is_bidirectional  ": true
31      }},
32      {{
33        "name": "eagle−grass ",
34        " description ": "eagle is  flying  through the sky",
```

```
35        " spatial_relation ": {{"above": 1.0, "below": 0.0, "in front of": 0.0, "behind": 0.0," left of": 0.0, " right of": 0.0}},
36        " importance_to_ask_score ": 0.1,
37        "name_entity_1 ":" eagle ",
38        "name_entity_2 ": "grass ",
39        " is_bidirectional ": false
40      }},
41      {{
42        "name": " tree −grass ",
43        " description ": "",
44        " spatial_relation ": {{"above": 0.5, "below": 0.0, "in front of": 0.0, "behind": 0.0, " left of": 0.25, " right of": 0.25}},
45        " importance_to_ask_score ": 0.1,
46        "name_entity_1 ":" tree ",
47        "name_entity_2 ": "grass ",
48        " is_bidirectional ": false
49      }},
50
51      ... [[a few additional examples]] ...
52
53   ]
54
55  Identify relationships between entities given the input given below. Strictly stick to the format.
56  Input: {{
57    "user_prompt": "{user_prompt}",
58    " entity ": "{entity_names}"
59  }}
60  Output:
```

## E.8. Verbalization Prompt Instruction

```
1  The chat history is as follows:
2  question: {action. verbalized_action } and answer: { observation }.
3  Turn the question and action into a single declarative sentence that describes the answer – do not phrase it as a question. Example output: the firetruck in the image is red.
```

## E.9. Merge Prompt Prompt Instruction

```
1  You are writing a prompt for a text−to−image model based on user feedback. The original prompt is {prompt}. The user has provided some additional information: { additional_info }.
     Please write a new prompt for the text−to−image model. The new prompt should be a meaningful sentence or a paragraph that combines the original prompt and the additional
     information. Do not add any new information that is not mentioned in the prompt or the additional information. Make sure the information in the original prompt is not
     changed. Make sure the additional information is included in the new prompt. Make sure the new prompt is a description of an image. If the additional information or the
     original prompt specifically says that a thing does not exist in the image, you should make sure the new prompt mentions that this thing does not exist in the image. DO
     NOT generate rationale or anything that is not part of a description of the image.
```

## E.10. $AICQ_{base}$ Prompt Instruction

```
1  ... [[ Instruction for the first question ]] ...
2
3  The original prompt was: { self. original_prompt } – Based on the original prompt please provide a question to ask about the image. The question should be as concise and direct as
     possible. The question should aim to learn more about the attributes and contents of the image, the objects, the spatial layout, and the style. Make sure that you question
     the answer within <question> and </question> markers
4
5  ... [[ Instruction for the following question ]] ...
6
7  Based on the chat history please provide a new question to ask about the image. the chat history is as follows and is enclosed in <chat_history> and </chat_history> markers:{self.
     chat_history } </chat_history> The question should be as concise and direct as possible. The question should aim to learn more about the attributes and contents of the image,
     the objects, the spatial layout, and the style. Make sure that you question the answer within <question> and </question> markers.'
```

## E.11. $AICQ_B$ Prompt Instruction

```
1  You are an intelligent agent that helps users generate images. Before generating the image requested by the user, you should ask the most important clarification questions to make
     sure you understand the key features of the image.
2  The user describes the image as: {user_prompt}.
3  The following is your belief of what the image contains, including the entities, attributes of each entity and relations between entities.
4  Each entity has "name", " descriptions ", "importance to ask score" and " probability of appearing". "Name" is the identifier of the entity. "Descriptions" is the description of the
     entity . "Importance to ask score" is how important it is for the agent to ask whether the entity exists. Probability of appearing" is the probability the agent estimated
     that this entity exits in the image.
5
6  Each entity has a list of attributes. Each attribute has "name", "importance to ask score" and "candidates ". "Name" is the identifier of the attribute. "Importance to ask score" is
     how important it is to ask about the exact value for the attribute of the entity. "Candidates" is a list of possible values for the attribute.
7
8  Each candidate value has a probability that describes how likely this candidate value should be assigned to the attribute.
9  For example, " Attribute Name: color, Importance to ask Score: 0.9, Candidates: [white: 0.5, black: 0.5]" means the color is either white or black, each with 0.5 probability. If you
     ask about attributes, you should ask about the attribute with the highest uncertainty. Your uncertainty can be judged by the probabilities. If the probabilities are 0.5 and
     0.5, you are uncertain. If the probabilities are 0.1 and 0.9, you are fairly certain.
10
11 The agent belief is:
12 { belief_state . __str__ ()}
13
14 Based on the user prompt "{user_prompt}" and the belief of the agent, please provide a question to ask about the image. The question should be as concise and direct as possible.
     The question should aim to obtain the most information about the style, entities, attributes, spatial layout and other contents of the image. Remember to ask for information
     that are critical to knowing the critical details of the image that is important to the user. The question should reduce your uncertainty about the user intent as much as
     possible. DO NOT ask question that can be answered by common sense. DO NOT ask question that are obvious to answer based on the user prompt "{user_prompt}". DO NOT ask any
     question about information present in the following user−agent dialogue within <dialogue> and </dialogue> markers.
```

```
15
16   <dialogue>
17   {conversation}
18   </dialogue>
19
20   DO NOT ask any question that has been asked in the dialogue above.
21
22   Your question does not have to be entirely decided by the belief. You can construct any question that make yourself more confident about what the image is.
23   Think step by step and reason about your uncertainty of the image to generate. Make sure to ask only one question. Make sure it is not very difficult for the user to answer. For
        example, do not ask a very very long question, which can take the user a long time to read and answer.
24   Make sure that you question the answer within <question> and </question> markers.
```

### E.12. HSA Question Prompt Instruction

```
1    You are constructing a text−to−image (T2I) prompt and want more details from the user.
2    You have to ask a question about the the most important entity or the attribute of the most important entity.
3    We have entity types: (i) explicit: directly ask question with options; (ii) implicit: ask whether this entity required for the image with yes or no as options; (iii) background:
        ignore the attribute value and directly ask the value of the entity. (iv) relation: add keyword like 'relation' to emphasize this entity is a relation.
4    Construct a simple question that directly asks this information from the user and also provides option that the user can pick from. Ask only one question and follow it with options
        .
5
6
7    Example1:
8    entity: rabbit
9    attribute: color
10   candidates: black, white, brown
11   entity_type: explicit
12   question: What color of the rabbit do you have in mind? a. black, b. white, c. brown. d. unkown. If none of these options, what color of the rabbit do you have in mind?
13
14   ... [[a few additional examples]] ...
15
16   Example5:
17   entity: $entity
18   attribute: $attribute
19   candidates: $candidates
20   entity_type: $entity_type
21   question:
```

## F. Automatic Evaluation

In Algorithm 2, we show the user-agent self-play procedures that we used to perform all automated evaluation.

### F.1. DSG Evaluations

The DSG (Cho et al., 2024) metric is used to compare the ground truth caption and the generated caption. DSG is adapted to parse text prompts into Davidsonian scene graph using the released code. The plots in Figure 11 show the T2T DSG metrics per turn. The results in Figure 11 show that significant gains in performance come from using proactive multi-turn agents. The blue line on the graph represents the simplest baseline which directly uses a T2I model and performs no updates to the initial prompt p0. Note that all multi-turn agents far exceed the baseline T2I model on all benchmarks for the DSG metric.

### F.2. Mitigation of T2I Errors

A limitation of the proposed pipeline is that its overall capability is constrained by the prompt-following abilities of the employed text-to-image (T2I) model. The proposed agent prototypes call off-the-shelf T2I models directly, treating T2I APIs as tools. This offers the benefit of seamlessly switching to better T2I models when they become available. In an effort to mitigate T2I errors caused by the model not following the prompt, we perform an experiment demonstrating that using agent-user QA pairs can improve T2I fidelity across a batch of N seeds.

The experiment design is as follows: Each QA pair from the agent-user dialogue is converted into a (yes/no) VQA question concerned with a single detail of the image. Then, using a VQA score metric with these new questions, we can remove erroneous images from a set of N seeds by filtering out images with low VQA scores. We perform this experiment on the DesignBench image-caption dataset.

The specific implementation is as follows:

1. Use the 30 ground truth (GT) prompts of DesignBench and generate 10 images from 10 different random seeds with Imagen.
2. Take average DINO (I2I) score for all images against GT image, this was found to be 0.7637.
3. Take the first 5 Q-A pairs from Ag2 turn each into a binary yes or no question for which the correct answer is yes.

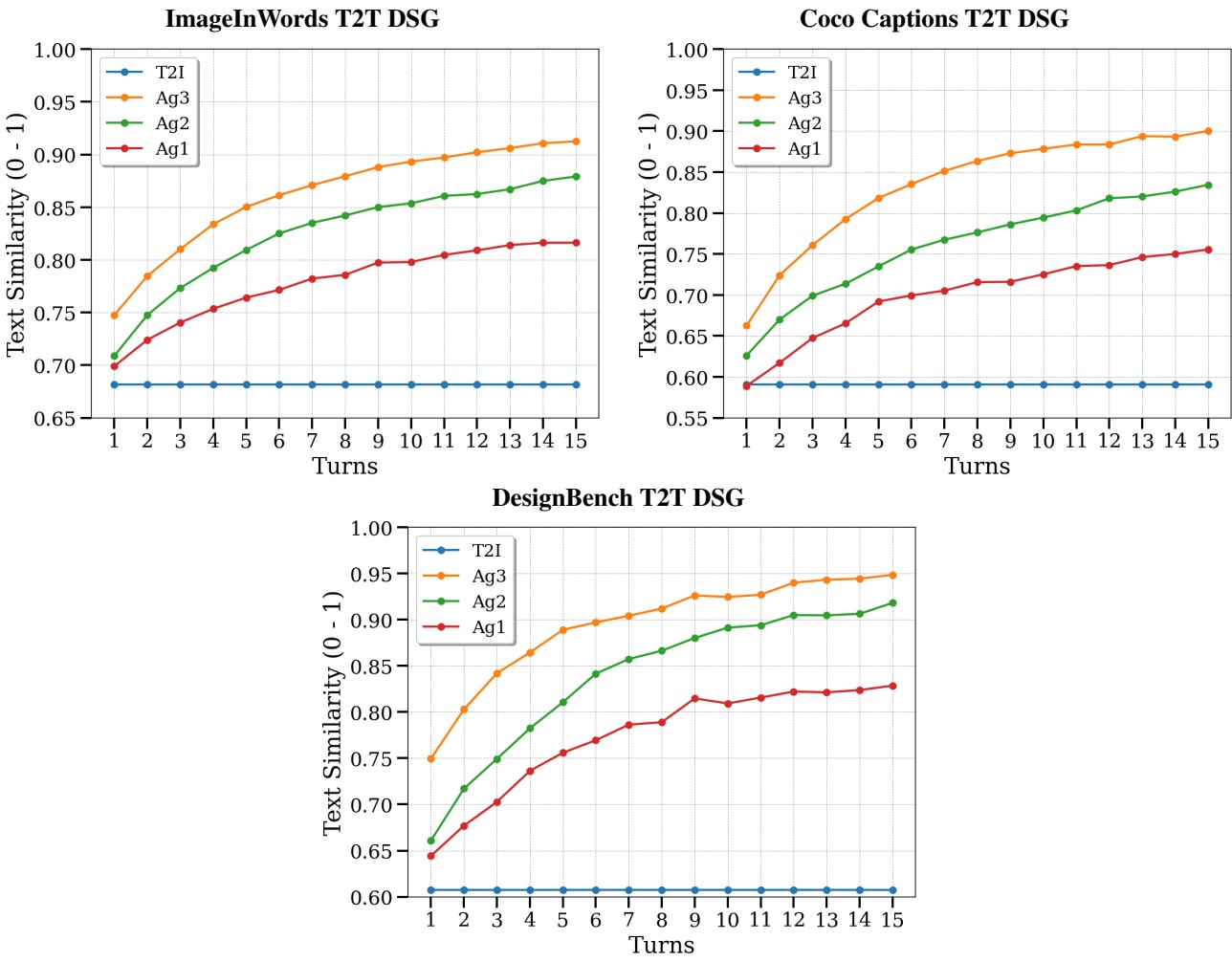

*Figure 11.* DSG score comparison between ground truth prompt and agent generated prompt reported at each turn. The performance of all agents increase with increase in number of turns.

4. Run the VQA scorer over all 10 images per caption.
5. Choose best image of the ten by selecting the image with highest score.
6. Take average DINO (I2I) score for best image against GT image: this was found to be 0.7838.
7. Calculate the $\delta$ between before and after filtering out images via agent QA pairs: $\delta = +.02$

In conclusion, this experiment demonstrates that by using the QA pairs from the agent in combination with the VQA score, we can improve image fidelity by filtering out images that do not follow the prompt. T2I models do not always follow a prompt exactly. They can make small errors or ignore a single detail while retaining all others. This is an inherent bottleneck of our agents; however, our results show that using the QA pairs from the agent-user interaction history allows us to overcome this limitation.

## G. Details on the Agent Interface

Below is a showcase of how users could interact with the belief graph and clarifications in a hypothesised interface, to better iterate their inputs, to reach a higher quality and satisfaction of outputs. This is a crudely hypothesised, intentionally simple interface for the sake of research, but could be iterated and improved upon in many ways depending on application and users.

---

**Algorithm 2** User-Agent Self-Play Algorithm

---

1: **Input:** Initial prompt $p_0$, User $u$, Agent $a$ (with $p_0$), $max\_turns$
2: **Output:** Refined prompt $p_f$
3: $p_f \leftarrow p_0$
4: **for** $turn\_id = 0$ **to** $max\_turns - 1$ **do**
5:     $action \leftarrow a.\text{SelectAction}()$
6:     $question \leftarrow a.\text{VerbalizeAction}(action)$
7:     $answer \leftarrow u.\text{AnswerQuestion}(question)$
8:     $a.\text{Transition}(action, answer)$
9:     $p_f \leftarrow a.prompt$
10: **end for**
11: **return** $p_f$

---

**1. Default state**    On load of the app, there would be a text prompt input and space for output images, as is common across typical T2I interfaces. There would also be space for the user to view either clarifications from the model, or a graph interface, as part of the overall "input" section as these would act as a further input for future model output iterations. See Figure 12 below as reference.

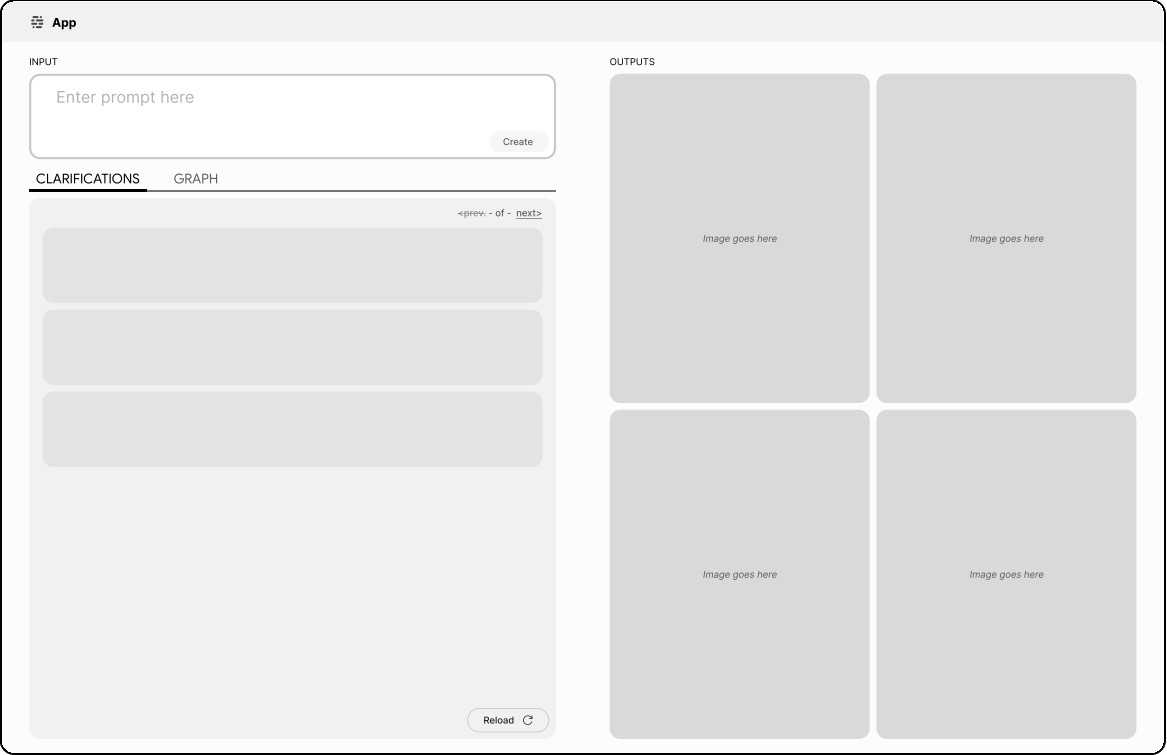

*Figure 12.* Default state of a possible interface.

**2. Output images, with Clarifications**    Once the user has submitted the prompt and the model has responded, there would be a set of images, as initial outputs from the users prompt. Below the input prompt would be a set of "Clarifications" in its populated state. These clarifications would ask the user specific questions that would be necessary to increase the specificity of the prompt, for the model to get a more accurate results aligned to the users intention, or to help the user realise their intention. Options would be given of the highest probability options for each Clarification, but the user could also fill in a totally new option via a free text field. Once answered by selection or text input, the clarifications would be added to the above, primary prompt for regeneration when the user selects. See Figure 13 below as reference.

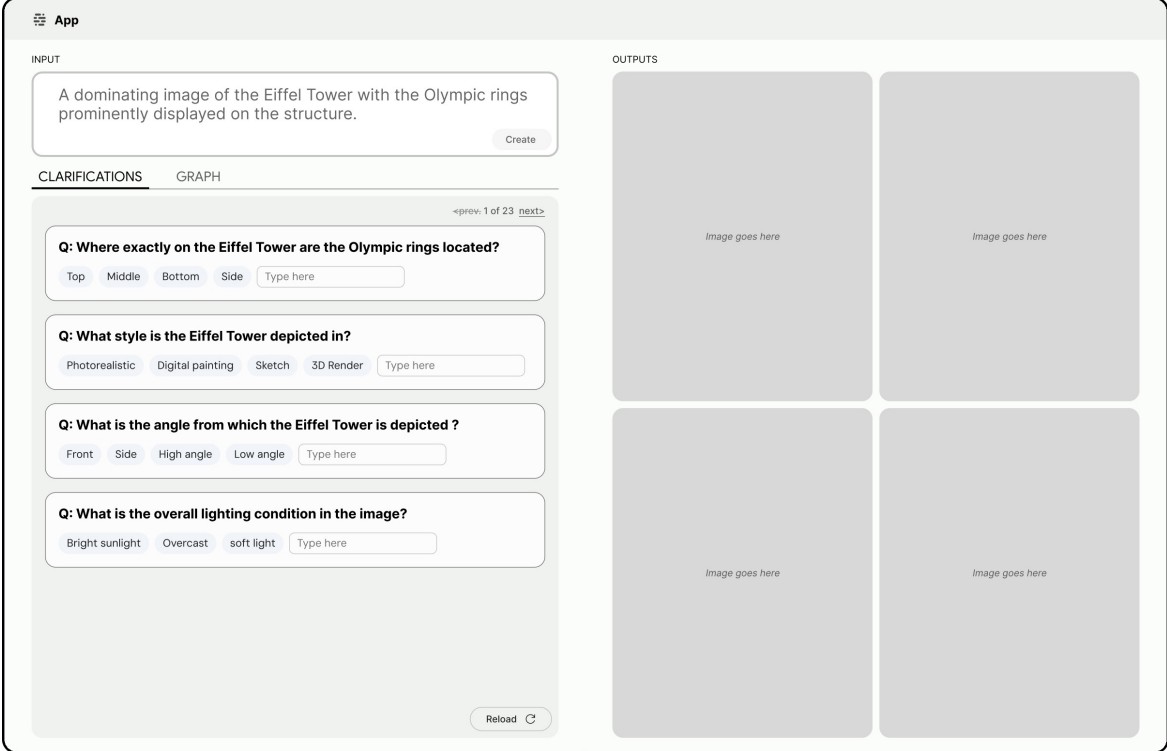

*Figure 13.* Interface once prompt has been input with clarifications.

**3. Graph Entities & Attributes**   Instead of the clarifications, the user could select to instead view a Graph by clicking Tab above the clarifications themselves. This graph would be populated with all Entities from the prompt explicit and implicit visually defined differently (in this diagram by the dotted line surrounds implicit entities, but is a filled line when surrounding explicit entities). The graph layout will be structured, depicting relationships concentrically i.e. "on", "in" or "under" for example, will become child entities, and be displayed within the parent entities' boundary. For example a 'Mug' that has the relationship of 'on' a 'Table' entity, will sit within the boundary of 'Table', as also would a 'Plate' if that had the same child-parent relationship.

Below the Graph would also be a list of 'cards' (i.e. boxed groups of information), one for each "explicit" or "implicit" entity. Within each card a user could see the status of implicit / explicit, and change this status to confirm or deny its presence. The user could also see a list of "attributes" associated to that entity, which the model has assumed. Each of these attributes could be changed by interacting with a list of alternatives via drop down. These lists are determined in terms of which items and order of items, based on the probability by which the model sees them, ordered with highest first. This probability would be made clear to the user to define the order by seeing the percentage next to the label. See Figure 14 below as reference.

**4. Graph Relationships**   The user would also be able to change the state of the Graph and Cards, to instead focus on the relationships between entities, by toggling to "Relations". In this state the user would be able to focus on two specific entities (e.g. 'mug' and 'table'), see the description of the relationship (e.g. 'the mug is sitting on the table') and if desired change the relationship to an alternative (e.g. 'on', changed to 'under') via a drop down of options which the model determined as alternative options ordered by probability, as per attributes. See Figure Figure 15 below as reference.

Once any of these changes are made the user could initiate a regeneration via the updated prompt to create a new set of output images, which can then be further refined via the same method.

# H. Details on User Studies for the Agent Interface

Below we describe the exact guideline definitions we shared with the user for a user study.

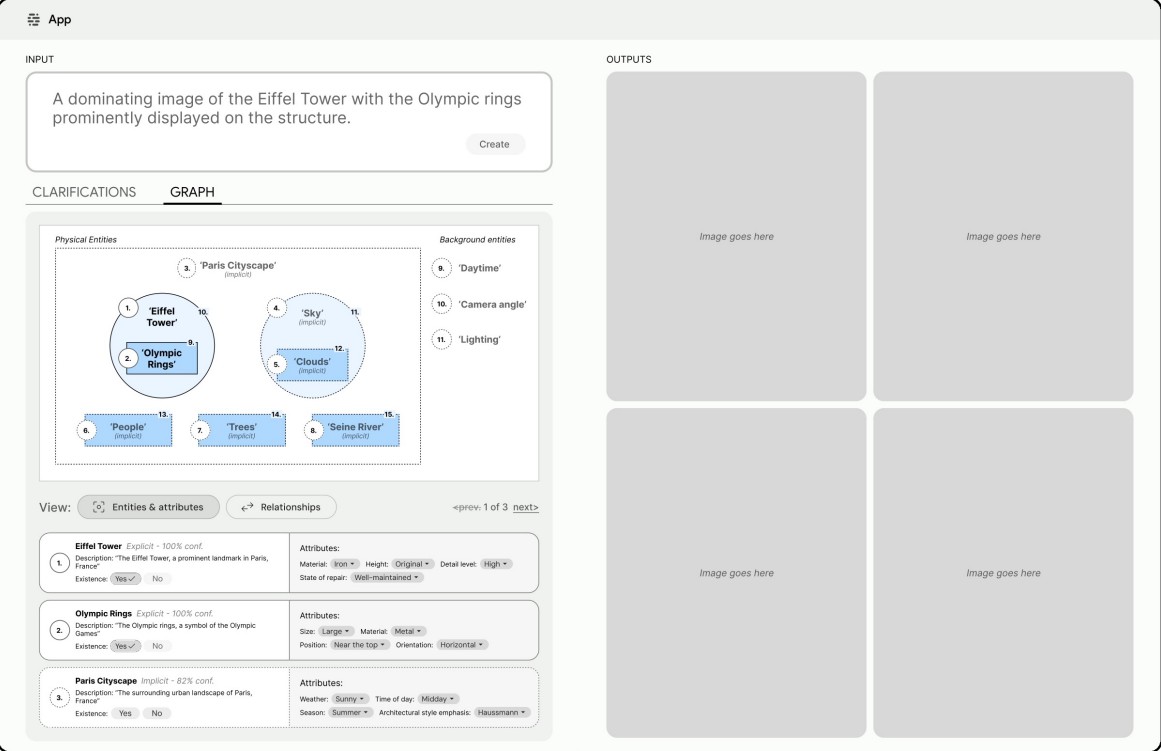

*Figure 14.* Interface with Graph displaying Entities, with cards below enabling a user to change attributes associated to each entity.

### H.1. Hypothesized Frustrations

We presented participants with the following hypothesized frustrations related to T2I model usage:

1. **Prompt Misinterpretation:** The model misunderstands complex relationships between entities in the input prompt.

2. **Many Prompt Iterations:** The model does not immediately generate what the user intends, requiring numerous iterative changes to the input prompt.

3. **Inconsistent Generations:** The model reinterprets the input prompt differently between iterations, causing unwanted changes in the generated images.

4. **Incorrect Assumptions:** The model makes incorrect assumptions or no assumptions when encountering gaps in the details provided in the input prompt, leading to undesired outputs.

Explanations of terms were given to users of:

1. "Entities" are single items that are intended to be in the image e.g. "Cat" and "Ball", from "make a sketch of a Cat playing with a Ball"

2. "Prompt" means the text written to communicate the intended output image e.g. the sentence "make a sketch of a Cat playing with a Ball" is the "Prompt", also known as "Input"

3. "Iterations" are each set of different image outputs by the model, taken from a different input, or even the same input just regenerated

The question asked for each Frustration were: "Please score the below frustrations (or issues) that could be related to Text to Image AI Generation"." Rank in terms of how much they relate to your current usage, with your most commonly used model or app."

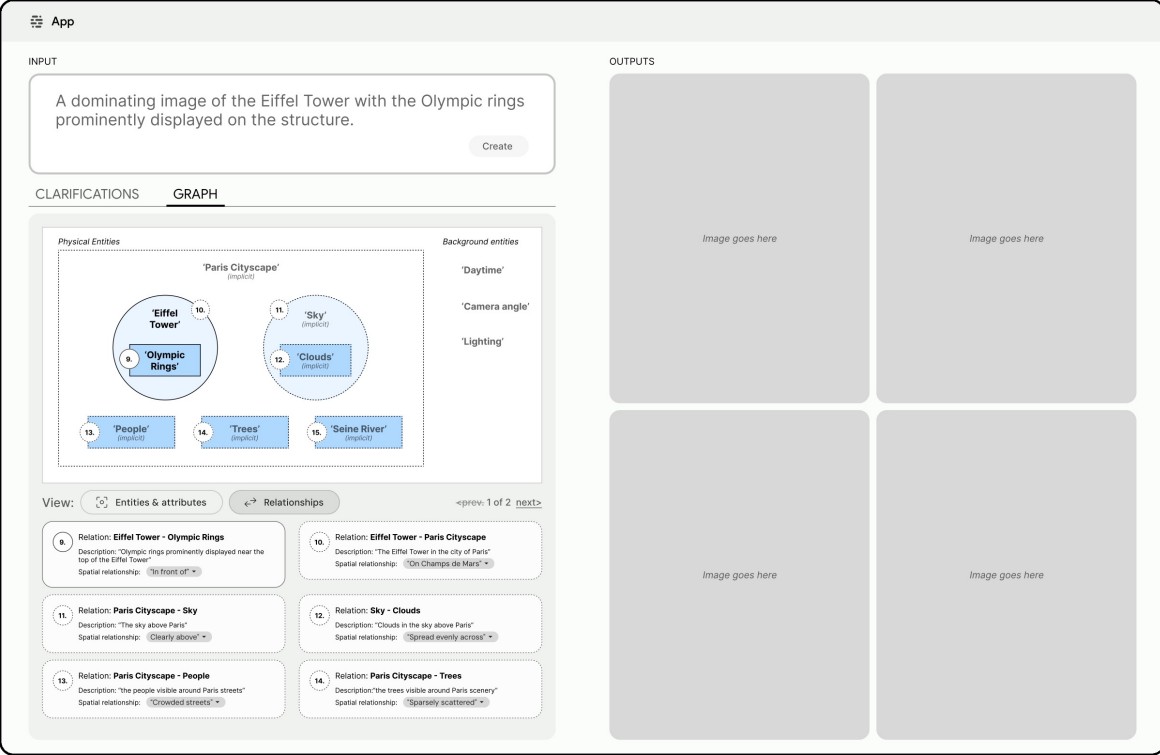

*Figure 15.* Interface with Graph displaying relations between Entities, with cards below enabling a user to change relationships between entities.

### H.2. Hypothesized Features

We proposed the following features as potential solutions to address the identified frustrations:

1. **Clarifications:** The model would ask specific clarifying questions about uncertainties in the prompt. These details would then be incorporated into subsequent iterations. For example: "Is the cat playing with: 1. a ball of wool, or 2. a tennis ball?"

2. **Graph of Prompt Entities:** A visual representation of all entities in the prompt as a graph, allowing users to see and edit attributes of each entity. E.g., seeing that the model has assigned "round," "small," and "wooden" as attributes to "table" and allowing the user to change them to "square" and "metal."

3. **Graph of Prompt Relationships:** A visual representation of relationships between entities in the prompt, allowing users to see and edit these relationships. E.g., seeing that "donut" is "next to" "coffee" and allowing the user to change the relationship to "on top of."

The questions asked for each feature were:

1. "How likely this feature is to help your current workflow if you had it now?". With response options of: "Very unlikely to help", "Unlikely to help", "Could help", "Likely to help", "Very likely to help".

2. "How soon would this feature deliver value to your work?" with response options of: "Very soon / immediately", "Sometime, "Not very soon".

Image references were given for each Feature as listed out below:

1. **Clarifications:**

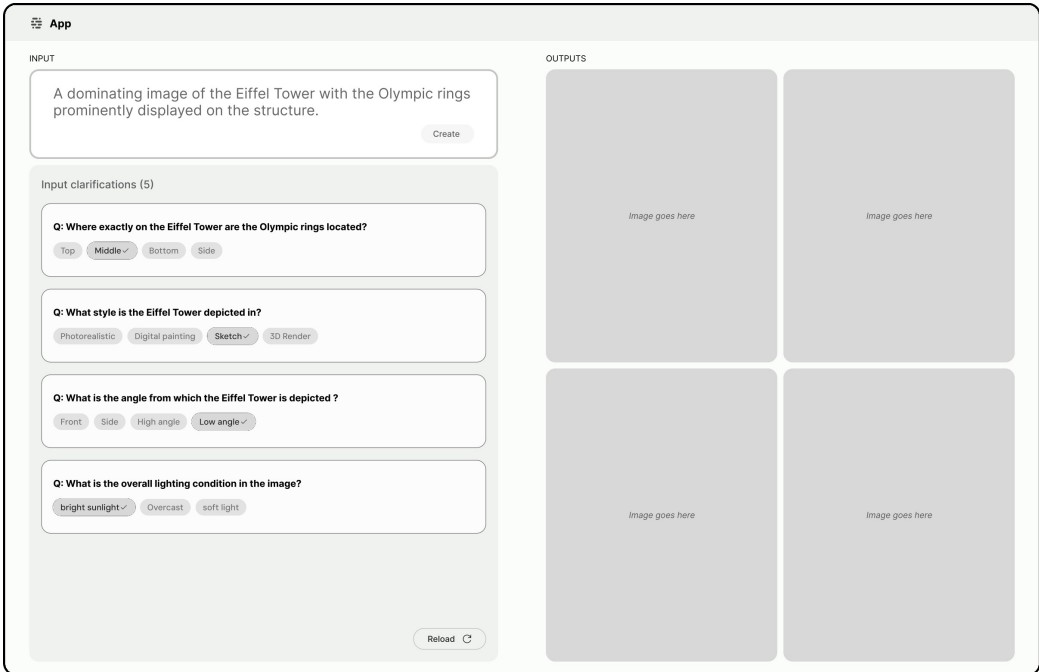

*Figure 16.* Stimulus image in the survey to test the Model clarifications feature.

2. **Graph of Prompt Entities:**

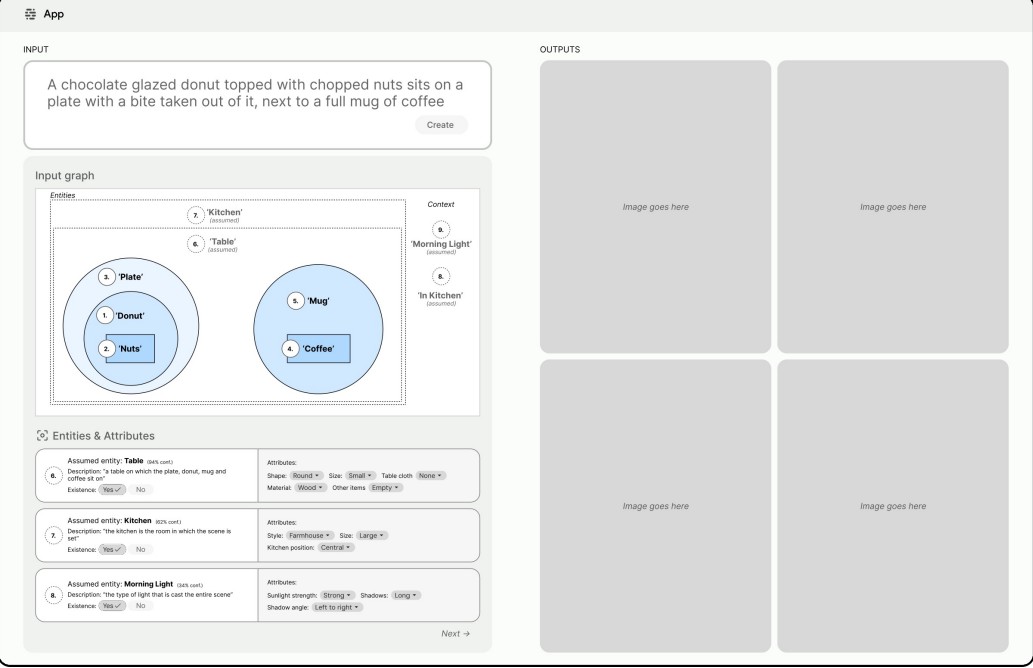

*Figure 17.* Stimulus image in the survey to test the Model Graph of Entities and Attributes feature.

3. **Graph of Prompt Relationships:**

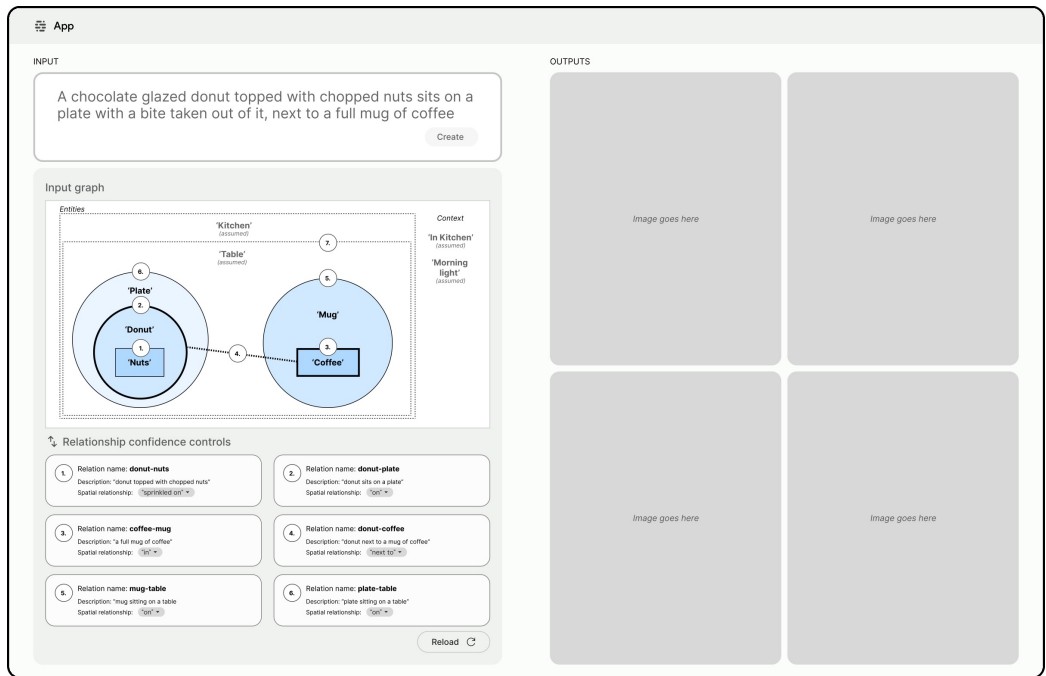

*Figure 18.* Stimulus image in the survey to test the Model Graph of Entity Relations feature.

## H.3. Human Study Results

Table 3 details the T2I usage frequency of the human subjects. Table 4 shows the percentage of human subjects that reported different kinds of frustrations in their experience of using T2I. Table 5 summarizes the results on expected speed of value delivered from different features of our agent prototypes. These results highlight the impact of our contributions.

*Table 3.* Breakdown of the T2I usage frequency of the 143 participants recorded

| Usage Frequency | No. of participants | (%) |
|---|---|---|
| Many times a day | 13 | 9.1 |
| Many times a week | 44 | 30.8 |
| At least once a week | 36 | 25.2 |
| At least once a month | 50 | 35.0 |

*Table 4.* Reported User Frustrations with existing T2I processes (% of participants)

| Frustration | V. Freq. (%) | Freq. (%) | Occas. (%) | V. Occas. (%) | No Issue (%) |
|---|---|---|---|---|---|
| Prompt Misinterpret. | 7 | 19.6 | 43.4 | 23.1 | 7 |
| Many Iterations | 10.5 | 44.8 | 28 | 11.9 | 4.9 |
| Inconsistent Gen. | 11.2 | 20.3 | 39.9 | 21 | 7.7 |
| Incorrect Assumptions | 7 | 23.1 | 39.2 | 20.3 | 10.5 |

# I. Details on Recruitment and Participant Consent

In this human user study participants were recruited with the following requirements:

1. Situated in North America and fluent in English.

*Table 5.* Expected speed of value delivered from features (% of users)

| Feature | Very soon / immediately (%) | Sometime(%) | Not very soon. (%) |
|---|---|---|---|
| Clarifications | 57.7 | 37.2 | 5.1 |
| Entity Graph | 49.6 | 34.8 | 15.6 |
| Relation Graph | 41.8 | 44 | 14.2 |

2. An age of 18 or above.
3. Have experience (i.e. at least once in the past month) using Text to Image AI Generation tools (e.g. Mid Journey, Stable Diffusion, ImageFX, DALL·E, Adobe Firefly etc.), for any purpose (work, non-work, just for fun).
4. The survey must be completed on a desktop computer - not a mobile phone.
5. Participants were recruited in North America and were compensated based on industry standards.

An internal review of the human-study experiment was first created to determine potential risks and harm to participants and found it to be low risk and we were not collecting any personally identifiable information. The review found that there were two potential risks. The first was the risk of a study with *experimental technology*: T2I AI models are still under development and may produce inaccurate, biased, or potentially offensive content. Participants must use caution when relying on the model's output and always exercise critical thinking. The second risk regarded *potential biases*. The model may reflect the biases present in the data it was trained on. These biases can manifest in various ways, such as favoring certain groups or stereotypes. By using this model, participants acknowledge that they have read and understand this disclaimer. They also agree to use the model responsibly and ethically.

In response to the findings of the internal review, risk mitigation was performed by serving generated images rather than having participants generate images themselves using the model. All image-prompt pairs and all generated images shown to users were pre-approved as a way to reduce risk and remove any overtly harmful or offensive content. Additionally a disclaimer and self consent form were placed at the beginning of the human-study survey to let participants know the type of questions they would be answering.

**Template of Human Rater Task 1: Evaluation of Issues in Individual Questions**

## Example Dialogue Analysis:

**Instructions**: Pretend you are in the following scenario - you are asking an AI model to create an image you have in mind, this image is displayed on the left. You first prompt the model with a short non-detailed description of the image, this description is called "original prompt" and is below the goal image on the left.

The model proceeds to ask you various questions to understand the specificities of the image you are trying to create. The dialogue between you and the AI model is shown in the middle column. You will go through the dialog turn by turn and answer the same rater questions about each turn so just focus on the highlighted turn rather than the entire dialog.

Your job in this task is to rate the clarity, soundness and efficiency of the highlighted question in the dialogue. This question was asked by the AI model in order to generate an image similar to the one you had in mind. Provide your rating by answering the questions in the rightmost column.

| Goal Image | Human - AI dialogue | Rater Questions |
|---|---|---|
| 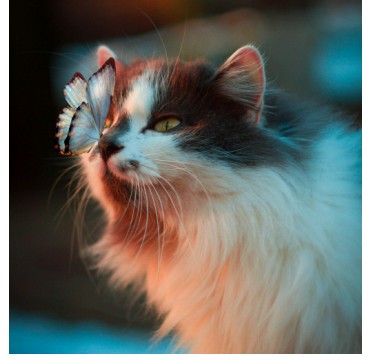 **Original Prompt** | **Q1.** *Which way is the cat facing?*

**A1.** *To the left.*

**Q2.** *What type of butterfly is on the cat?*

**A2.** *Not sure what type but it is a blue butterfly with black around the edges of the wings.*

**Q3.** *Where is the butterfly positioned on the cat's face?* | Select one or more issues for this particular turn:

[ ] The question is completely unclear and ambiguous

[ ] The question is irrelevant to the goal image and prompt

[ ] The question is not helpful; doesn't gain any new information from what was previously asked |

*Figure 19.* An example of the template presented to human raters. Human raters are asked to mark any issues a question contains that could pose a disturbance to the user. Approximately 8k questions per Agent are rated. The results are shown in Figure 4.

**Template of Human Rater Task 2: Evaluation of Similarity between the generated image and the Human-AI dialog and the original prompt.**

## Example Image Analysis:

**Instructions**: Pretend you are in the following scenario - you are asking an AI model to create an image you have in mind, this image is displayed on the left. You first prompt the model with a short non-detailed description of the image, this description is called "original prompt" and is below the goal image on the left.

The model proceeds to ask various questions to understand the specificities of the image you are trying to create. The dialogue between you and the AI model is shown in the middle column.

Based on the original prompt and the human and AI dialogue - the AI model produces a final image shown in the 3rd column.

Your job in this task is to rate the produced image based on how well it fits the goal image and separately how well it fits the prompt and the dialogue. Do this by answering the questions in the rightmost column.

| Goal Image | Human – AI dialogue | Produced Image | Rater Questions |
|---|---|---|---|
| 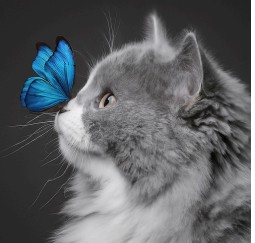  **Original Prompt**  `A fluffy gray and white cat with a butterfly on its face.` | **Q1.** *Which way is the cat facing?*  **A1.** *To the left.*  **Q2.** *What type of butterfly is on the cat?*  **A2.** *Not sure what type but it is a blue butterfly with black around the edges of the wings.*  **Q3.** *Where is the butterfly positioned on the cat's face?*  **A3.** *On the cat's nose.* | | **Key = 1:** Very Close, **2:** Fairly close some fair differences, **3:** Equally close and different **4:** Fairly different some similarities, **5:** Very different no similarities  Rank how well the produced image corresponds to the original prompt and the dialogue? ( 1 - 5 Scale) |

*Figure 20.* An example of the template presented to human raters. Human raters are asked to rank the correspondence of each image to the agent-user dialog and original prompt. Approximately 1.5k image-dialog pairs are rated using 3 human raters. Results in Figure 4.

**Template of Human Rater Task 3: Evaluation of the generated image similarity to the goal image.**

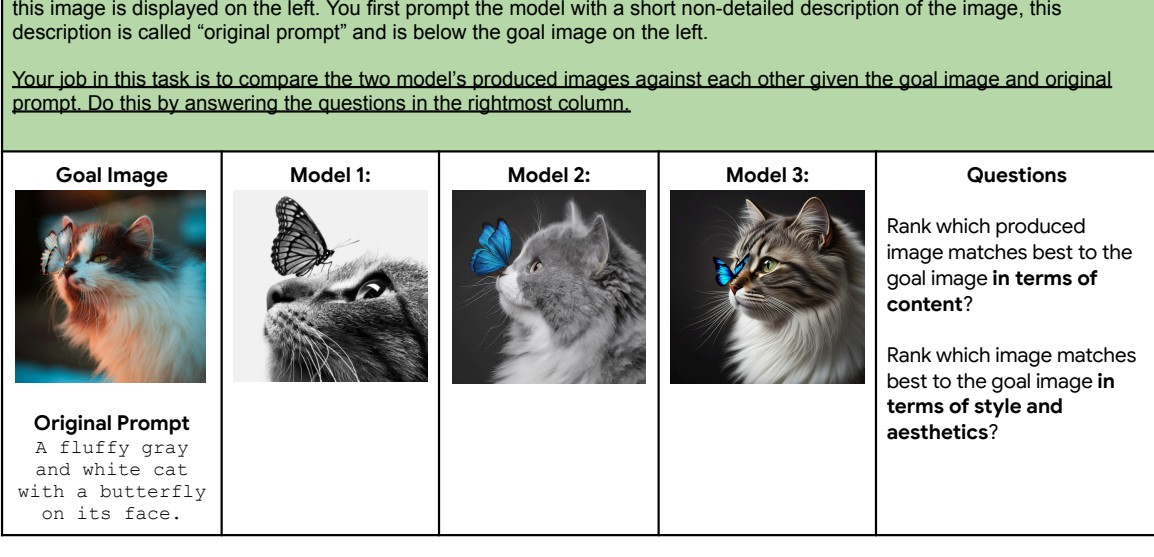

*Figure 21.* A template of the task presented to human raters. Human raters are asked to rate the images produced by the three proposed multi-turn agents and a single-turn T2I model against a ground truth image for which the original prompt was derived and the answers to the agents questions were derived. Approximately 550 image-dialog pairs (250 from ImageInWords, 250 from COCO-Captions and 21 from DesignBench) per agent are rated using 3 human raters. The generated images were presented in a random order and were unlabeled and the human rater was tasked with ranking the images from best to worst. The results from the study are shown in Table 1.

