# OpenReview forum: "Proactive Agents for Multi-Turn Text-to-Image Generation Under Uncertainty"
_ICML.cc/2025/Conference — ICML 2025 poster_

### Official Review · Reviewer_dRg9 · 2025-03-01

**Overall Recommendation:** 3

**Summary:**

This paper focuses on the information-missing issue of existing AI applications, specifically image generation in the paper. Instead of passively waiting for humans to revise the prompt, this paper proposes a proactive design such that the agent system could proactively interact with the users to get the missing information. This paper has five main contributions:  (1) the introduction of belief graphs for uncertainty modeling; (2) a proactive T2I agent prototype; (3) an evaluation pipeline; (4) a benchmark; and (5) extensive experiments.

**Claims And Evidence:**

Yes

**Essential References Not Discussed:**

N/A

**Experimental Designs Or Analyses:**

Yes, I checked the dataset, experiment results, and case studies.

**Methods And Evaluation Criteria:**

Yes

**Other Comments Or Suggestions:**

The bottom lines of all tables are missing. Better to add them.

**Other Strengths And Weaknesses:**

Strength:
1. A proactive agent is a sound solution for solving the information-missing problem and might be an essential step of future agent design.
2. The paper is clearly written and easy to follow.
3. The experiments and analysis is comprehensive

Limitations:
1. Some detailed design choices require more justifications; for details, please refer to the questions for the authors section.
2. The efficiency might be a limitation for the proactive agent system to be useful in real applications.

**Questions For Authors:**

1. The current brief graph requires heavy human design, which will limit the generalization of the proactive agent system to other domains other than the image generation task. How do you prevent that?
2. In real application, how do you guarantee that the users will clearly answer the questions raised by the system?
3. This proactive design will introduce an extra efficiency issue. How do you prevent that?

**Relation To Broader Scientific Literature:**

The proactive agent design has been used in the general agent design but not in the image generation scenario, to the best of my knowledge.

**Theoretical Claims:**

N/A, no theoretical claim

---

> ### Author Rebuttal · Authors · 2025-04-01
>
> > The current brief graph requires heavy human design, which will limit the generalization of the proactive agent system to other domains other than the image generation task. How do you prevent that?
>
> Generating the belief graph only requires some few-shot examples, which are not difficult to write. Writing those examples offers humans the opportunity to insert their expert knowledge, so that the LLM’s behavior can be controlled. Moreover, for new domains, it is sometimes not necessary to re-design the belief graph parsing approach. We tried the same belief parsing approach without modifying anything for creative writing tasks, and the belief graph we obtained from the models is reasonably good, e.g., it is able to identify the characters, their attributes like clothing, super power etc and relations between characters. One future direction to further lower the effort of human design is to perform meta learning, where we give the model examples of different domains, and ask it to generalize to new domains.
>
> > In real application, how do you guarantee that the users will clearly answer the questions raised by the system?
>
> We cannot “guarantee” the user behavior but the agents can guide users. The questions the agents ask often contain several options for the answer. For example, the agent may ask “What is the color of the rabbit? (a) white, (b) grey, (c) brown, (d) mixed colors.” The user can choose an option (this guarantees that the answer is clear), or answer with words directly.
>
> > This proactive design will introduce an extra efficiency issue. The efficiency might be a limitation for the proactive agent system to be useful in real applications. How do you prevent that?
>
> We can use parallelism and selective generation to prevent efficiency issues. The agent is a system that operates in multi-threads/multi-processes. In the agent prototypes we developed, the efficiency of belief parsing was significantly improved by generating attributes and relations for different entities in parallel. The T2I models can be called while the next belief state and action selection are in progress. Our framework also supports the development of more sophisticated agents in real applications, which can incorporate strategies to selectively generate partial belief states that are important for showing to the users. This can be especially useful if the belief state is very large.
>
>
> We will fix the formatting of tables.

---

### Official Review · Reviewer_tR3z · 2025-03-10

**Overall Recommendation:** 3

**Summary:**

This paper addresses the challenge of underspecified user prompts in text-to-image (T2I) generation by introducing proactive agents capable of multi-turn interactions. These agents actively seek clarification through targeted questions and utilize a "belief graph" to represent and refine their understanding of user intent. The proposed approach aims to bridge the gap between user expectations and model outputs. Empirical evaluations show that the method achieves a VQAScore twice as high and is rated as helpful by 90% of human participants.

**Claims And Evidence:**

The claims made in the submission supported by clear and convincing evidence.

**Essential References Not Discussed:**

N/A

**Experimental Designs Or Analyses:**

The analysis consists of two parts: the VQAScore and human opinions. It can be considered sound.

**Methods And Evaluation Criteria:**

The proposed method focuses on utilizing an editable belief graph for human-AI interaction. The idea is simple yet seems reasonable and effective.

**Other Comments Or Suggestions:**

The writing of the paper needs improvement. Please try to avoid redundant wording and excessively long sentences. This issue appears even in the abstract, e.g., "As a result, users **commonly** have to **painstakingly** and **repeatedly** refine their prompts." This writing style makes the paper difficult to read.

**Other Strengths And Weaknesses:**

Strengths:

- Important problem and practical value: The problem of underspecified prompts is common and significant in user-AI interactions. A common user without prompt-engineering training always finds it hard to convey their intentions efficiently to the models, causing frustration. This method effectively addresses the common issue of vague or ambiguous prompts, leading to more accurate and user-aligned image generation.

- Effectiveness: The authors employ both human studies and automated evaluations to assess the effectiveness of their approach. Notably, over 90% of human subjects found the proactive agents and belief graphs beneficial to their T2I workflow. Additionally, the agents achieved at least twice the VQAScore compared to standard single-turn T2I generation.

Weaknesses:
- The universal prompt design: It seems like the design of the prompt lacks consideration of the alignment and prompt-following capabilities of the text-to-image model. From my personal user experience with T2I models, the correct wording and structure of the prompt may also greatly influence the quality of the generated image.

**Questions For Authors:**

See weaknesses.

**Relation To Broader Scientific Literature:**

N/A

**Theoretical Claims:**

The paper have no theoretical claims.

---

> ### Author Rebuttal · Authors · 2025-04-01
>
> > The universal prompt design: It seems like the design of the prompt lacks consideration of the alignment and prompt-following capabilities of the text-to-image model. From my personal user experience with T2I models, the correct wording and structure of the prompt may also greatly influence the quality of the generated image.
>
> We have added a new method to take into account the T2I alignment of T2I models. Please see more descriptions in Q1 rebuttal to Reviewer F62f
>
> > writing
>
> Thank you for pointing these out. We will rewrite the long sentences and make the writing more accessible.

---

### Official Review · Reviewer_CQhi · 2025-03-11

**Overall Recommendation:** 3

**Summary:**

This paper proposed a proactive text-to-image agent designed to mitigate the issue of uncertain prompts by allowing users to do multi-turn interactions. The key contributions of the paper include:

1. Belief Graphs – A structured representation of model uncertainty, allowing users to visualize and edit entities, attributes, and relationships.
2. Proactive Questioning – The agent actively seeks clarifications from users to refine image generation.
3. New Benchmark (DesignBench) – A dataset created to test agent performance in complex scenes with both long and short descriptions.

The study demonstrates that multi-turn agents significantly outperform standard T2I models in both automated and human evaluations, achieving at least twice the VQA score within five interaction turns.

## Update after rebuttal

The authors have addressed all of my concerns. I vote to accept the paper and appreciate their detailed and thoughtful responses.

**Claims And Evidence:**

1. Claim: Multi-turn interaction improves T2I alignment

    Evidence: Table 1, Fig. 2 and 3 demonstrate that multi-turn agents outperform single-turn T2I models across various datasets, with at least a 2x improvement in VQAScore.

2. Claim: Belief graphs effectively communicate and refine uncertainty

    Evidence: Over 85% of human participants found belief graphs helpful.

3. Claim: Proactive questioning enhances user experience

    Evidence: Human evaluations indicate that 90% of participants expect interactive clarifications to be beneficial.

Issue: The main contribution of this work is the proposal of the belief graph and multi-turn interaction with users. However, the comparison includes only one baseline focused on single-turn generation. I suggest expanding the comparison to include more baselines, such as Recraft V3 or Midjourney v6, which have been reported to outperform Imagen 3 in numerical reasoning cases in the Imagen 3 paper. This would provide a more comprehensive evaluation of general performance improvements and adaptability compared to current state-of-the-art methods.

**Essential References Not Discussed:**

None

**Experimental Designs Or Analyses:**

- The experiments are well-structured, covering both automated and human evaluations.
- The use of DINOv2 embeddings for image similarity and VQAScore for text-to-image alignment are reasonable choices.

**Methods And Evaluation Criteria:**

Yes, the paper employs 2 evaluation methods for evaluating how the generated image align with user’s intent (prompt)

1. Automated Evaluation – Uses a simulated user to converse with a T2I agent
2. Human Studies – Participants rate the efficacy of the proposed framework with human subjects

**Other Comments Or Suggestions:**

None

**Other Strengths And Weaknesses:**

- Strengths
    1. The idea to proactively clarify user input is novel and helpful for efficiently generate a target image.
    2. Leveraging LLM as a user to provide clarification on a pre-given image is an efficient strategy.
- Weaknesses
    1. The description of the three agents is difficult to understand in terms of their differences, and the information is scattered across too many sections, including Section 4.3, Supp. C, and E. I suggest improving the clarity of the descriptions of the three agents in the main paper.

**Questions For Authors:**

- The motivation for including Agents 1 and 2 in the comparison is unclear.
   - Why are Agent 1 and Agent 2 designed in the current setting?
   - Additionally, why should Agent 1 ask whether the entity "cake" is present in the image, as shown in Fig. 5 of the supplement? Does this occur before or after generating the first image?
- Does the performance of direct LLM prompting—where an image is given, the LLM describes it as a prompt, and a T2I model generates an image in a single turn—outperform multi-turn iteration?
- How does the belief graph scale to complex scenes with dozens of entities?
   - Are there practical limitations on the number of entities and relationships that can be handled effectively?
- To what extent does the T2I model adhere to the belief graph representation?
- If the generated image does not follow the belief graph and the clarification question fails to identify the issue, how can the user provide feedback to help the agent improve the generated image?

**Relation To Broader Scientific Literature:**

This work contributes to clarify the prompt before generation which would enhance the efficiency for trial and error on prompt engineering

**Theoretical Claims:**

The paper does not propose new theoretical results but extend concepts from belief state representations.

---

> ### Author Rebuttal · Authors · 2025-04-01
>
> > baseline
>
> We used only one single-turn T2I baseline because we want to keep the T2I backbone consistent between the baseline and all agents. To ensure the consistency, we will run all agent experiments with a different model (such as the suggested ones) and we can definitely add this comparison to the paper.
>
>
> > the clarity of the descriptions of the three agents
>
> Please see Q3 in the rebuttal for Reviewer F62f.
>
> > Why are Agent 1 and Agent 2 designed in the current setting?
>
> We designed the two agents to compare rule-based approach and LLM-based question-asking without explicit rules. Ag1’s action selection is purely rule-based: it maximizes the approximated information gain of questions (weighted by importance scores in the belief graph) and the model can only ask about the existence of an entity, the attribute of any entity or the relation between any pairs of entities in the belief graph. Ag2 also relies on the belief graph but it does not compute any heuristic scores and instead puts the belief graph in the context to prompt an LLM to ask questions.
>
> > why should Agent 1 ask whether the entity "cake" is present in the image, as shown in Fig. 5 of the supplement? Does this occur before or after generating the first image?
>
> Because Ag1 is rule-based and checks both the probabilities and importance scores, it may try to confirm the existence of an entity the user mentioned, because the importance score for that entity is very high and the probability is less than 1. The reason that the probability is less than 1 is because sometimes, when an entity is mentioned, the image doesn’t necessarily have to include the entire entity. Moreover, the belief graph parsing is not perfect, so errors can occur since Ag1’s question purely depends on the belief graph. In our current implementation, this occurs before generating the images.
>
> > Does the performance of direct LLM prompting—where an image is given, the LLM describes it as a prompt, and a T2I model generates an image in a single turn—outperform multi-turn iteration?
>
> If the LLM describes the image in detail and the T2I model has good text-image alignment, it should outperform the multi-turn approach. This is because the multi-turn approach’s goal is to eventually collect all information about the image, and the detailed description is the ground truth of what information should be collected.
>
> > How does the belief graph scale to complex scenes with dozens of entities? Are there practical limitations on the number of entities and relationships that can be handled effectively?
>
> The belief graph can scale to dozens of entities as long as the belief graph parsing prompts and every entity together with their attributes in the belief graph fits the context length and generation length limits of the LLM. Practical limitations: as shown in Algorithm 1, the time complexity of belief parsing is linear in the number of entities and the number of relations, but in practice, we make parallel calls to the LLM to generate attributes of entities and relations in parallel. So the time complexity is limited by the requests per second the LLM can afford and how many parallel threads / processes the python environment can afford.
>
> > To what extent does the T2I model adhere to the belief graph representation?
>
> The belief graph describes the state of the agent, not the T2I model. The belief graph entails a distribution over possible prompts with different combinations of entities, attributes and relations. The distribution over the images the agent generates will adhere to the belief graph relatively faithfully if the T2I model’s T2I alignment is good. We also describe an approach to further enhance the alignment between agent behaviors and user intents in Q1 rebuttal to Reviewer F62f.
>
> > If the generated image does not follow the belief graph and the clarification question fails to identify the issue, how can the user provide feedback to help the agent improve the generated image?
>
> The user can modify the current prompt the agent has and inspect/modify the belief graph to ensure it aligns with the user intent. Admittedly, the T2I models may still fail to generate images that align with the text descriptions. We described an approach to further enhance the alignment in Q1 rebuttal to Reviewer F62f. It is still possible that the VLM might not be able to tell whether the answers to questions conditioned on the generated image match with the user answers. These are intrinsic limitations of the current T2I models or VLMs, and improving those models are out of the scope of this paper. However, if there exist improved T2I models and VLMs, we can seamlessly integrate them into our agents.

---

> > ### Comment · Reviewer_CQhi · 2025-04-03
> >
> > Most of my questions have been addressed.
> >
> > - Minor thoughts regarding Ag1:
> > I understand the current logic, but I still wonder: if the prompt already specifies an entity (e.g., a cake), does Ag1 really need to confirm its existence before generation? This step feels somewhat redundant. Would it be more useful for Ag1 to ask questions that surface uncertainties instead of reaffirming what the prompt already states?
> >
> > > The belief graph describes the state of the agent, not the T2I model.
> >
> > - My understanding is that the belief graph reflects what the T2I model should generate. So my original question was about this: To what extent does the T2I model actually conform to the belief graph in practice? I believe this is clarified in the next point.
> >
> > > We described an approach to further enhance the alignment in Q1 rebuttal to Reviewer F62f.
> >
> > - While this approach is promising, I would caution that VQA-based evaluations have known limitations, as discussed in VQAScore [A]. These evaluations may struggle with complex prompts such as "someone talks on the phone happily while another person sits angrily." In such cases, divide-and-conquer methods like Davidsonian [B] tend to generate nonsensical questions (e.g., “is the someone happy?” or “is there another person?”), which raises concerns about their reliability in complex scenarios.
> >
> > Overall, I remain positive about the work and would keep my current rating.
> >
> > [A] Lin et al., VQAScore: Evaluating Text-to-Visual Generation with Image-to-Text Generation. ECCV 2024
> >
> > [B] Cho et al., Davidsonian Scene Graph: Improving Reliability in Fine-Grained Evaluation for Text-Image Generation. ICLR 2024

---

> > > ### Author Response · Authors · 2025-04-08
> > >
> > > > Minor thoughts regarding Ag1
> > >
> > > Ag1's question generation process, as described in Appendix E.2, relies on the MHIS (Most Important to Ask Score) strategy. This strategy leverages the importance scores and probabilities associated with entities, attributes, and relations in the belief graph, and constructs heuristic scores for possible questions. The heuristic score dynamically determines the usefulness of posing a question about a specific element.
> > >
> > > Roughly speaking, the heuristic score of entity-existence-confirmation questions is higher for entities that have high importance score, high entropy in its attributes, and high probability of existence (but less than 1). The rationale behind this design is that entities that are very likely to exist should be clarified first. This design was based on our trial and errors.
> > >
> > > For an entity that is explicitly mentioned in the prompt, the probability of existence can be estimated to be 0.99 and this is still considered less than 1. So based on the suboptimal heuristic strategy, this approach may, at times, result in the formulation of redundant inquiries.
> > >
> > > > an approach to further enhance the alignment in Q1 rebuttal to Reviewer F62f.... known limitations in VQA-based evaluations
> > >
> > > Thank you for the advice. The approach we adopt uses the questions in the history of agent-user interaction, and this can alleviate the nonsensical question problem in the Davidsonian scene graph approach.
> > >
> > > > To what extent does the T2I model actually conform to the belief graph & more on T2I alignment
> > >
> > > Thank you for clarifying the question. The extent to which the T2I model conform to the belief graph is currently bounded by the image-prompt alignment capabilities of the T2I model.
> > >
> > > In an effort to mitigate images that contain T2I errors - we perform and show a new experiment that finds using the agent-user QA pairs can improve T2I fidelity over a batch of N seeds.
> > >
> > > Each QA pair from agent-user dialogue is converted into a (yes/no) VQA question concerned about a single detail of the image. Then using VQA score metric with the new questions - we can remove erroneous images from a set of N seeds, by filtering out images with low VQA scores.  We perform this experiment on the DesignBench image-caption dataset from the paper.
> > >
> > > The design of the experiment is as follows
> > > - Using the 30 ground truth (GT) prompts of design bench, generate 10 images from 10 different random seeds with Imagen.
> > >     - Take average DINO (I2I) score for all images against GT image:this was found to be  0.7637.
> > > - Take the first 5 Q-A pairs from Ag2 and change it to a yes or no question that should have a yes answer.
> > >     - Run the VQA scorer over all 10 images per caption.
> > >     - Choose best image of ten (a.k.a. image with highest score).
> > >     - Take average  DINO (I2I) score for best image against GT image: this was found to be 0.7838.
> > >     - Find the Delta between before and after filtering out images via agent QA pairs: Delta is +.02.
> > >
> > >
> > > Conclusions of the experiment: we find that by using the QA pairs from the agent in combination with the VQA score we can improve image fidelity by filtering out images which do not follow the prompt. T2I models do not always follow a prompt exactly. They can make small errors or ignore a single detail while retaining all others. This is an inherent bottleneck of our current pipeline, however we show using the QA pairs from the pipeline that we can overcome this limitation.

---

### Official Review · Reviewer_F62f · 2025-03-14

**Overall Recommendation:** 3

**Summary:**

This paper addresses the issue of suboptimal image generation caused by vague or incomplete prompts provided by users to text-to-image (T2I) generators. It introduces proactive T2I agents designed to improve image generation by actively asking users clarification questions. These agents maintain their understanding in the form of a belief state, which users can view and modify directly. Additionally, the paper presents a scalable, automated evaluation benchmark for assessing T2I systems. Experimental results indicate that 90% of human participants found the proactive agents and editable belief states beneficial, with the proposed approach significantly enhancing the quality of generated images, as evidenced by higher VQAScores.

## update after rebuttal
The reviewer still recommends weak acceptance.

**Claims And Evidence:**

* Claim: Proactive T2I agents improve user interaction
* Claim: Belief state interpretability and editability

These two claims are supported by the experimental result indicating that 90% of human participants found proactive agents and the belief state useful, providing clear and convincing evidence.

* Claim: Improvement in image alignment

The work shows improved quality of generated images, supported convincingly by reported significant increases in metrics such as VQAScores.

**Essential References Not Discussed:**

Asking for clarifications for text-to-image generation is not a new idea. The work missed this related work [1] that also investigated into the ambiguity in user's prompt and asked the user for clarifications.

[1] Is the Elephant Flying? Resolving Ambiguities in Text-to-Image Generative Models. https://arxiv.org/abs/2211.12503

**Experimental Designs Or Analyses:**

The overall experiment design is sound and well-defined. However, there are some concerns in the design of the agents in the prposed work:

Ag2 and Ag3 use LLMs without vision input, which indicates that the exploration is only conducted in the textual space. This might cause misalignment between the textual exploration and the image generation.

**Methods And Evaluation Criteria:**

The evaluation protocol that guides the proposed DesignBench, as discussed in Sec 5.1, is sound and well-defined. The proposed DesignBench offers a good example of how to evaluate the performance of T2I systems with proactive agents.

**Other Comments Or Suggestions:**

One minor detail: the bottom lines of Table 1 and Table 2 are missing.

**Other Strengths And Weaknesses:**

The writing is a bit unclear in the explanation of each agent. The reader needs to look at the appendix to understand the design of each agent.

**Questions For Authors:**

Is the capability of the whole pipeline limited by the prompt-following capabilities of the text-to-image model (i.e., if given a prompt that has a lot of details from clarifications, does the model always generate an image that is aligned with the prompt)?

**Relation To Broader Scientific Literature:**

The proposed work improves the text-to-image workflow by introducing proactive agents that can ask questions to the user to improve the quality of the generated images. The work builds on prior text-to-image generation works and focuses on a novel aspect of it.

**Theoretical Claims:**

No theoretical claims are made in the paper.

---

> ### Author Rebuttal · Authors · 2025-04-01
>
> > Q1: “Is the capability of the whole pipeline limited by the prompt-following capabilities of the text-to-image model?” Also in Experimental Designs Or Analyses: “misalignment between the textual exploration and the image generation.”
>
> The current agent prototypes call off-the-self T2I models directly, i.e., treating T2I API as tools. The benefit is that it is seamless to switch to better T2I models when they become available.
>
> We have now included a new module under our agent framework to further improve text-image alignment with vision feedback. The key idea is to use the existing <agent question, user answer> history and ask a VLM those agent questions on the generated image. If the VLM’s answers do not match the corresponding user answers, the agent can refine the prompt (such as adding “ensure the color of the rabbit is grey”; this refinement component can be replaced with better prompt optimization methods) and call T2I models to re-generate the image. We are testing this approach and will include more results in the updated paper.
>
>
> > Q2: Related work
>
> Thank you for sharing the related work. This paper proposes to use clarification questions to resolve ambiguities (a prompt has multiple meanings), but we aim to use clarification questions to resolve underspecification (the prompt is not ambiguous, but lacks information to fully describe the image). We will include this paper in the related work and note the difference.
>
> > Q3: Clarity of agent descriptions.
>
> Thank you for pointing this out. The main paper has page constraints and we had to make certain tradeoffs. The core of this paper is the new framework and overall design of proactive agents and automated evaluation. All 3 agents share the implementation of belief parsing (Algorithm 1), belief transition (Section 4.1) and principles of asking questions to collect information (Section 4.2). Those are detailed in the main paper. However, we recognize that readers may also be curious about the specific implementations. We will consolidate the relevant sections in appendix and add the important details back to the paper, including (1) Ag1 selects actions by maximizing the approximated information gain of questions, so that the questions can reduce the entropy of belief graphs; (2) Ag2 uses in-context learning and generates questions conditioned on user prompt, belief graph and conversation history; (3) Ag3 uses the LLM to generate questions conditioned on the principles of asking questions and conversation history.
>
> We will fix the formatting of tables.

---

> > ### Comment · Reviewer_F62f · 2025-04-03
> >
> > The authors have addressed my questions in the review. The authors are encouraged to add the details mentioned in the rebuttal to the camera-ready version of the paper.

---

### Decision · Program_Chairs · 2025-05-01

**Decision:**

Accept (poster)

**Comment:**

This paper received 4 reviews and 2 ethics reviews. The reviewers found the problem tackled in the paper relevant (tR3z), the idea presented sound (dRg9), the claims well supported (F62f, CQhi, tR3z, dRg9), and the evaluation well designed and overall convincing (F62f, CQhi, dRg9). The reviews raised concerns about:
1) the paper's presentation, in particular, the clarity of the agents description and motivation (F62f,CQhi, tR3z);
2) the experiments, which included only one T2I model (CQhi);
3) some design choices, which did not appear adequately justified (F62f, dRg9) - e.g. the exploration being conducted in the text space;
4) the design, which disregarded the alignment and prompt-following capabilities of the T2I model (tR3z, F62f).

The reviews also highlighted some missing related work (F62f), and questioned the efficiency of the approach (dRg9).
The ethics reviews raised concerns about the lack of details on participants, their concern, and the overall recruiting process (DJcV, sCzK).

The authors' response addressed most of the reviewers' questions, comments and concerns. In particular, the response clarified the concerns related to presentation by answering the reviewers' questions in detail, adequately discussed the design choices questioned by the reviewers, and explained how to increase the efficiency of the approach. The response also positioned the paper w.r.t. the missing reference. The authors also summarized the review/response process for the ACs.

The AC takes the authors' message into consideration when discussing with the reviewers and making the recommendation. After discussion, there is consensus that the paper addresses an important problem with high practical value, and that the proposed approach is sound and well validated. Therefore, the AC recommends to accept and encourages the authors to add the clarifications requested by the reviewers as well as the information requested in the ethics reviews (participants recruitment and consent) to the camera-ready version of the paper.